

# Twenty-first century Southern Hemisphere impacts of ozone recovery and climate change from the stratosphere to the ocean

Ioana Ivanciu[1], Katja Matthes[1], Arne Biastoch[1,2], Sebastian Wahl[1], and Jan Harlaß[1]

[1]GEOMAR Helmholtz Centre for Ocean Research Kiel, Kiel, Germany
[2]Faculty of Mathematics and Natural Sciences, Christian-Albrechts Universität zu Kiel, Kiel, Germany

**Correspondence:** Ioana Ivanciu (iivanciu@geomar.de)

**Abstract.** Changes in stratospheric ozone concentrations and increasing concentrations of greenhouse gases (GHGs) alter the temperature structure of the atmosphere and drive changes in the atmospheric and oceanic circulation. We systematically investigate the impacts of ozone recovery and increasing GHGs on the atmospheric and oceanic circulation in the Southern Hemisphere during the twenty-first century using a unique coupled ocean-atmosphere climate model with interactive ozone chemistry and enhanced oceanic resolution. We use the high emission scenario SSP5-8.5 for GHGs under which the springtime Antarctic total column ozone returns to 1980s levels by 2048 in our model, warming the lower stratosphere and strengthening the stratospheric westerly winds. Novel results of this study include the springtime stratospheric circulation response to GHGs, which is characterized by changes of opposing sign over the Eastern and Western Hemispheres, the opposing responses of the Agulhas leakage to ozone recovery and increasing GHGs, and large uncertainties in the prediction of atmospheric and oceanic circulation changes related to whether the ozone field is prescribed or calculated interactively.

By performing a thorough spatial analysis of the predicted changes in the stratospheric dynamics, we find that the GHG effect during spring exhibits a strong dipole pattern, which contrasts the GHG effect during the rest of the year and which was previously not reported, as it cannot be detected when zonal means are considered. Over the Western Hemisphere, GHGs drive a warming of the lower stratosphere and a weakening of the westerlies, while over the Eastern Hemisphere they drive a cooling and a strengthening of the westerlies. Associated with these changes, planetary waves break higher up in the stratosphere over the Eastern Hemisphere, strengthening the polar downwelling and inducing dynamical warming in the upper stratosphere, while weakening the downwelling and inducing dynamical cooling in the lower stratosphere. The opposite occurs over the Western Hemisphere. Because the changes in the Western Hemisphere dominate during November in our model, we find that during this month GHGs lead to a weakening of the lower branch of the Brewer-Dobson Circulation, reinforcing the weakening caused by ozone recovery.

At the surface, the westerly winds weaken and shift equatorward due to ozone recovery, driving a weak decrease in the transport of the Antarctic Circumpolar Current and in the Agulhas leakage, which transports warm and saline waters from the Indian into the Atlantic Ocean. The increasing GHGs drive changes in the opposite direction that overwhelm the ozone effect. The total changes at the surface and in the oceanic circulation are nevertheless weaker in the presence of ozone recovery than those induced by GHGs alone, highlighting the importance of the Montreal Protocol in mitigating some of the impacts of climate change.





We additionally compare the combined effect of interactively calculated ozone recovery and increasing GHGs with their combined effect in an ensemble in which we prescribe the CMIP6 ozone field. This second ensemble simulates a weaker ozone effect in all the examined fields. The magnitude of the difference between the simulated changes at the surface and in the oceanic circulation in the two ensembles is as large as the ozone effect itself. This shows that the choice between prescribing or calculating the ozone field interactively can affect the prediction of changes not only in the atmospheric, but also in the oceanic circulation.

## 1 Introduction

Ozone depletion has been the major driver of change in the Southern Hemisphere (SH) atmospheric circulation during the last decades of the twentieth century (e.g., Polvani et al., 2011b). The effects of ozone depletion extend from the stratosphere (e.g., McLandress et al., 2010; Keeble et al., 2014), where the ozone hole is located, down to the surface and to the ocean circulation (e.g., Thompson et al., 2011; Previdi and Polvani, 2014). A decline in the absorption of shortwave (SW) radiation associated with ozone depletion in the past led to the cooling of the Antarctic lower stratosphere in austral spring, which in turn resulted in the acceleration of the stratospheric westerly winds (Thompson and Solomon, 2002; Gillett and Thompson, 2003) and a delay in the breakdown of the polar vortex (e.g., Waugh et al., 1999; Langematz et al., 2003; McLandress et al., 2010; Keeble et al., 2014). At the same time, the summer Brewer-Dobson circulation (BDC) in the SH has strengthened in response to changes in wave activity associated with the delayed breakdown of the polar vortex (Li et al., 2008, 2010; Oberländer-Hayn et al., 2015; Abalos et al., 2019). The impact of ozone depletion on the westerlies extends to the surface, where the mid-latitude jet strengthens and shifts poleward during the austral summer and the Southern Annular Mode (SAM) experiences a shift towards more positive values (Gillett and Thompson, 2003; Thompson and Solomon, 2002; Son et al., 2010; Thompson et al., 2011). The associated shift in the southern storm-track resulted in precipitation changes (Polvani et al., 2011b), extending to the subtropics (Kang et al., 2011). In addition, the edge of the Hadley cell shifted poleward (Son et al., 2010; Polvani et al., 2011b; Min and Son, 2013; Waugh et al., 2015), causing an expansion of the subtropical dry zones in the SH.

The impacts of ozone depletion are also felt by the ocean. The strengthening of the surface westerlies driven by ozone depletion and enhanced by climate change is tightly linked to an increase in the wind stress over the Southern Ocean. In response to the wind stress changes the northward Ekman transport across the Southern Ocean increased, leading to enhanced divergence (convergence) and upwelling (downwelling) to the south (north) of the maximum westerly wind stress, and therefore enhancing the ventilation of the Southern Ocean (e.g., Waugh et al., 2013). The Southern Ocean accounts for about 40% of the global oceanic carbon sink (e.g., Khatiwala et al., 2009) and there is evidence that ozone depletion weakened the uptake of $CO_2$ in this region, through its effect on the upwelling of carbon-rich deep waters (Le Quéré et al., 2007; Lenton et al., 2009). Both the increase in the Southern Ocean ventilation and the decrease in the oceanic $CO_2$ uptake are indicators of an enhanced Meridional Overturning Circulation (MOC). Earlier studies that used low resolution models indeed showed that the subpolar meridional overturning cell strengthens (e.g., Fyfe and Saenko, 2006; Sigmond et al., 2011), the Antarctic Circumpolar Current (ACC) shifts poleward (e.g., Oke and England, 2004) and its transport increases (e.g., Fyfe and Saenko, 2006; Sigmond et al.,



2011) in response to the poleward intensification of the wind stress over the Southern Ocean. However, another important consequence of the westerly wind stress strengthening is the intensification of eddy activity in the Southern Ocean (e.g., Morrison and Hogg, 2013; Hogg et al., 2015; Patara et al., 2016), which cannot be properly simulated in models that do not have a sufficiently high resolution. In high resolution models that resolve the oceanic mesoscale eddies, the enhanced eddy activity drives changes in the MOC that oppose the changes due to enhanced Ekman transport in a mechanisms known as

"eddy compensation" (e.g., Farneti et al., 2010; Viebahn and Eden, 2010). Therefore, the residual MOC changes are weaker than simulated by low resolution models, but the compensation is not perfect (Farneti et al., 2010; Morrison and Hogg, 2013; Munday et al., 2013). Similarly, while the changes in Ekman transport and Ekman pumping act to increase the tilt of the isopycnals across the ACC and therefore the ACC transport, the eddies offset these changes by flattening the isopycnals. This is known as "eddy saturation" and results in minimal changes in the ACC transport due to the increased wind stress (e.g.,

Farneti et al., 2010; Morrison and Hogg, 2013). Observations seem to support this mechanism, as past changes in temperature and salinity within the ACC resulted only in a deepening of the isopycnals, consistent with a poleward shift of the ACC, but not in a change in the meridional tilt of the isopycnals, implying that the ACC transport has not changed (Böning et al., 2008). This highlights the importance of using climate models that are capable of capturing the effects of eddies for studying the future changes in the Southern Ocean circulation, as it is the case of the model used in this study.

The strengthening of the westerlies due to ozone depletion also resulted in an increase in the wind stress curl over the southern part of Indian Ocean, which drives decadal changes in the Agulhas leakage (Rouault et al., 2009; Durgadoo et al., 2013). The Agulhas leakage represents the inter-ocean exchange of waters around the southern tip of Africa. Taking the form of rings, eddies and filaments that detach from the Agulhas Current as it loops back into the Indian Ocean, the Agulhas leakage transports warm and saline Indian Ocean waters into the South Atlantic. Once in the Atlantic, these waters enter the upper limb

of the Atlantic Meridional Overturning Circulation (AMOC) and are transported northwards (Gordon et al., 1992; Donners and Drijfhout, 2004). Ocean models driven by observational-based reanalysis at the surface simulate an increase in the Agulhas leakage in the second half of the twentieth century as a result of the changes in the westerly winds (Biastoch et al., 2009; Rouault et al., 2009; Schwarzkopf et al., 2019). This is confirmed by a positive trend in an observation-based reconstruction of Agulhas leakage (Biastoch et al., 2015). According to an estimate of leakage from satellite altimetry, this positive trend

appears to have ceased since the late twentieth century (Le Bars et al., 2014), likely related to the pause in surface westerlies trends reported by Banerjee et al. (2020). Variations in the leakage of Indian Ocean waters into the Atlantic have been linked to changes in the thermohaline properties of the Atlantic Ocean and to variations in the strength of the AMOC. Weijer et al. (2002) showed that, in a coarse resolution Ocean General Circulation Model with mixed boundary conditions allowing for the development of density anomalies, Agulhas heat and salt anomalies alter the overturning circulation in the Atlantic Ocean.

Coupled climate models simulate an advective pathway for salinity anomalies from the Agulhas region into the North Atlantic on timescales of 30-40 years, but their typical low resolution leads to an overestimation of the Agulhas leakage and salinity biases that cause the Agulhas leakage salinity anomalies to be too weak (Weijer and van Sebille, 2014). These results are nevertheless consistent with a high resolution ocean simulation in which an increased Agulhas leakage in response to changes in the surface westerlies leads to a salinification of thermocline waters spreading into the North Atlantic on time scales of two





to three decades (Biastoch and Böning, 2013). A more recent study with a high resolution ocean model that used a Lagrangian approach reported that the Agulhas leakage is the major source of waters for the upper limb transport of the AMOC, with an estimated transit time to the North Brazil Current at 6°S of about a decade (Rühs et al., 2019). Rühs et al. (2019) additionally found that Agulhas waters undergo a substantial modification of their thermohaline properties during their transit through the South Atlantic, resulting in a net density gain through a net salinity increase. The same study also showed that an increase in

Agulhas leakage results in an increased Agulhas contribution to the upper limb transport of the AMOC. The Agulhas leakage was also found to act as a source of decadal AMOC variability (Biastoch et al., 2008) and to be significantly correlated with the Atlantic Multidecadal Oscillation at a lag of 15 years (Biastoch et al., 2015). A shut down of the Agulhas leakage has been associated with a northward shift of the Intertropical Convergence Zone and consequent changes in precipitation, reduced equatorial upwelling, and a cooling and freshening of the Equatorial Undercurrent in the tropical Atlantic, as part of the initial

response simulated in a coupled climate model (Haarsma et al., 2011). Furthermore, paleoclimate studies suggest that the Agulhas leakage was low during glacial and high during inter-glacial periods and that it played an important role in glacial terminations through its influence on the AMOC (Knorr and Lohmann, 2003; Peeters et al., 2004). The Agulhas leakage is the main source of central water masses in the Benguela upwelling system, an important primary production region, and it has been hypothesized that a future increase in leakage could increase the contribution of Agulhas waters to the region and lead

to changes in the oxygen, $CO_2$ and nutrient content (Tim et al., 2018). An increased Agulhas leakage was additionally found to contribute to the warming of the tropical Atlantic (Lübbecke et al., 2015) and to the increase in the Atlantic Ocean heat content observed over the last decades (Lee et al., 2011; Biastoch et al., 2015). Given the manifold effects of variations in the Agulhas leakage, it is important to understand how the leakage will change in the twenty-first century under the influence of ozone recovery and increasing greenhouse gases (GHGs). The future behavior of the Agulhas leakage is investigated here for

the first time in a coupled climate model, as part of a broader analysis of the effects of ozone recovery and climate change.

Ozone depletion occurred during the last decades of the twentieth century as a result of anthropogenic emissions of ozone depleting substances (ODSs). ODSs are gases containing halogens, particularly chlorine and bromine, that take part in ozone-depleting chemical cycles in the stratosphere. The most severe ozone loss, the ozone hole, reoccurs every spring in the Antarctic lower stratosphere. There, cold conditions facilitate the formation of polar stratospheric clouds, which serve as surfaces for

efficient chlorine activation, resulting in strong ozone depletion. The 1987 Montreal Protocol on Substances that Deplete the Ozone Layer and its subsequent amendments and adjustments regulated the production of ODSs and mandated their phasing-out. As a result, the ozone in the stratosphere is expected to recover during the twenty-first century. The most recent estimates of Antarctic total column ozone (TCO) recovery obtained from the Chemistry-Climate Model Initiative (CCMI) ensemble indicate a return to 1980 levels in 2060, with a $1\sigma$ spread of 2055-2066, when the multi-model mean is considered (Dhomse

et al., 2018). When a weighted multi-model mean accounting for model independence and performance is used, the recovery is projected to occur in 2056, with a 95% confidence interval of 2052-2060 (Amos et al., 2020). Detecting ozone recovery in observations is complicated by large year-to-year variability (Chipperfield et al., 2017) and by the short record since the ODSs peaked in the 1990s. Trends in ozone since the year 2000 vary considerably with region and height (Chipperfield et al., 2017). Over Antarctica, observations show that ozone loss has leveled off and started to reverse since the beginning of the century



(Solomon et al., 2016, 2017; Banerjee et al., 2020). Solomon et al. (2016) identified a significant (at 90% confidence) increase in Antarctic TCO over 2000-2014 during September, a month with relatively low dynamical variability compared to the rest of the austral spring. Assuming compliance with the Montreal Protocol, ozone recovery should become more easily detectable in the next decades, as ODSs continue to decline. However, Montzka et al. (2018) reported that the rate of decline of CFC-11, an important ODS, slowed since 2012 and linked this to new emissions that violate international regulations. Therefore,

the success of the Montreal Protocol, which is considered a major achievement in terms of global environmental protection (Chipperfield et al., 2017), is contingent on continued global efforts to eliminate ODSs.

The recovery of the ozone hole is expected to largely reverse the effects of ozone loss. However, increasing levels of GHGs due anthropogenic activities also affect the atmospheric circulation (e.g., Kushner et al., 2001), as well as ozone recovery itself (Rosenfield et al., 2002; Waugh et al., 2009a; Eyring et al., 2010; Revell et al., 2012). Their influence is expected to oppose that

of ozone recovery in the twenty-first century. This complicates projections of future changes in the SH circulation. The current study aims to separately quantify the effects of ozone recovery and of increasing GHGs on the SH climate during the twenty-first century and to evaluate their relative importance under a high GHG emission scenario. We systematically investigate the future impacts of ozone recovery, increasing GHGs and their combined forcing from the SH stratosphere to the surface and extending to the Southern Ocean, providing for the first time a comprehensive view of the changes to the SH climate projected

to occur during the twenty-first century.

Although a number of previous studies employed single-forcing model simulations to explicitly separate the effects of ozone recovery and increasing GHGs on various aspects of the SH climate during the twenty-first century (Shindell and Schmidt, 2004; Perlwitz et al., 2008; Oman et al., 2009; Karpechko et al., 2010; McLandress et al., 2010, 2011; Polvani et al., 2011a; Oberländer et al., 2013; Polvani et al., 2018, 2019), many of them used older-generation models and forcing scenarios dating

back to the third or fifth phases of the Coupled Model Intercomparison Project (CMIP), and none of them investigated the changes in the Agulhas leakage and their impacts. Furthermore, with two exceptions (Polvani et al. (2018) and the two-part study by McLandress et al. (2010, 2011)), these studies used either atmospheric general circulation models or coupled ocean-atmosphere climate models that prescribe the ozone field (Shindell and Schmidt, 2004; Karpechko et al., 2010; Polvani et al., 2011a), chemistry-climate models (CCMs) that compute ozone chemistry interactively, but are not coupled to a dynamic ocean

(Oman et al., 2009; Oberländer et al., 2013; Polvani et al., 2019), or both types of models (Perlwitz et al., 2008). Prescribing the ozone field has been shown to alter the response of the SH circulation to ozone changes because of the temporal interpolation from monthly mean values to the model's time step (Neely et al., 2014) and the missing ozone asymmetries (Waugh et al., 2009b; Li et al., 2016) and ozone-radiative-dynamical feedbacks (Haase et al., 2020). In addition, before CMIP6, the prescribed ozone field was the same for all future GHG scenarios (e.g., Eyring et al., 2013). Therefore, the effect of GHGs on ozone was

not properly captured. However, the analyses by Morgenstern et al. (2014) and Chiodo and Polvani (2016) showed that the effect of GHGs on ozone is important as it offsets part of the direct influence of GHGs on the SH dynamics. The studies that used CCMs do not suffer from these deficiencies, but are limited by the specified sea surface temperature and therefore do not fully capture the climate response to ozone and GHG changes.





We revisit the results of the above-mentioned studies using a state-of-the-art coupled ocean-atmosphere chemistry climate
model forced by GHGs and ODSs according to the Shared Socioeconomic Pathway (SSP) 5-8.5 used in CMIP6 and we expand
these results in three ways. 1) We analyze spatial patterns of changes in several stratospheric fields in addition to the zonal
mean analysis customary in previous studies and we show that the response of the austral spring SH stratospheric dynamics
to increasing GHGs exhibits a marked zonal structure that has not been reported before. 2) We make use of a high resolution
(0.1°) ocean nest in the South Atlantic and western Indian Ocean that resolves the mesoscale features characteristic of the
Southern Ocean and the Agulhas System, and we separate for the first time the impacts of ozone recovery and of increasing
GHGs on the ocean circulation in these regions. Our model setup allows us to draw a clear connection between stratospheric
ozone changes and Southern Ocean circulation changes. 3) We compare our future SH climate projections from the interactive
chemistry version of the model with projections from the same model, but in the configuration that prescribes the CMIP6 ozone
field. This is motivated by the recent findings of Ivanciu et al. (2021), that the stratospheric circulation response to past ozone
changes is affected by the specification of the CMIP6 ozone field.

The study is structured as follows: Sect. 2 presents our model simulations and methodology, Sect. 3 gives an estimate of
ozone recovery in our model, Sect. 4 examines the impacts of ozone recovery and increasing GHGs on the atmospheric and
oceanic circulation in the SH and compares the combined impact in simulations with prescribed and interactive ozone, Sect. 5
discusses our results and Sect. 6 provides our summary.

## 2 Data and methodology

### 2.1 Model description and experimental design

We employ the state-of-the-art coupled climate model Flexible Ocean Climate Infrastructure (FOCI; Matthes et al., 2020). The
atmospheric component of FOCI is ECHAM6.3 (Stevens et al., 2013) in the T63L95 setting, corresponding to a horizontal
resolution of approximately 1.8° by 1.8° and 95 vertical hybrid sigma-pressure levels. The 95 vertical levels are chosen to
achieve a high vertical resolution, mainly in the stratosphere, to allow for a proper representation of the quasi-biennial oscil-
lation (Matthes et al., 2020). The model top at 0.01 hPa is located well above the stratopause, such that processes throughout
the entire stratosphere are simulated. Solar activity variations are included in FOCI following the recommendations of the
SOLARIS-HEPPA project (Matthes et al., 2017) for CMIP6. ECHAM6 is coupled to the Model for Ozone and Related Chemi-
cal Tracers (MOZART3; Kinnison et al., 2007), which simulates chemical processes in the atmosphere. 182 chemical reactions
with 52 chemical tracers focusing on stratospheric chemistry are simulated, including all relevant catalytic ozone loss cycles.
The ozone chemistry is therefore interactively simulated and the ozone field responds to changes in temperature, dynamics and
other chemical constituents such as GHGs. Land surface biogeophysical and biogeochemical processes are simulated using the
JSBACH land model version 3 (Brovkin et al., 2009; Reick et al., 2013).

The oceanic component of FOCI is the Nucleus for European Modelling of the Ocean version 3.6 (NEMO3.6; Madec
and the NEMO team, 2016), coupled to the LIM2 sea ice component (Fichefet and Maqueda, 1997). We use the ORCA05
configuration of NEMO, which has a global nominal resolution of 1/2° and 46 $z$ levels. Additionally, we take advantage of the





special nesting capabilities of FOCI to enhance the oceanic resolution in the South Atlantic and western Indian Ocean to 1/10°
using the INALT10X nest described by Schwarzkopf et al. (2019). A two-way nesting approach is used (Debreu et al., 2008),
whereby the nest receives information at its boundaries about the global ocean state from the ORCA05 host and it feeds back
its fine-scale state to the host at regular time intervals, prior to each coupling time step. Therefore, the impact of the nest is also
felt outside of the nest area. INALT10X resolves mesoscale features such as eddies in the Agulhas System region and, in part,
also in the Southern Ocean. The Agulhas leakage in INALT10X is therefore more realistic than that simulated in ORCA05
(Schwarzkopf et al., 2019). Outside of the nesting region, where the resolution does not allow for the eddies to be resolved,
an eddy parametrization scheme (Gent and Mcwilliams, 1990) is used, varying temporally and horizontally with the growth of
the baroclinic instability up to a maximum of 2000 $\mathrm{m^2\,s^{-1}}$ (Treguier et al., 1997).

FOCI was described in detail and was validated against observations by Matthes et al. (2020). The impacts of ozone depletion
and increasing GHGs on the atmospheric circulation in FOCI over the last decades of the twentieth century were investigated
by Ivanciu et al. (2021) and are in good agreement with those reported in previous studies.

Four ensembles were performed with FOCI for the twenty-first century. Each ensemble comprises of three members that
differ only in their initial conditions. The first ensemble, INTERACT_$O_3$, is forced using ODSs and GHGs according to the
SSP5-8.5, characterized by high GHG emissions. The $CO_2$ concentrations reach 1135 ppm and the radiative forcing reaches 8.5
W $\mathrm{m^{-2}}$ by 2100 in this scenario (Meinshausen et al., 2020). This ensemble is used to assess the combined effect of the increase
in GHGs and of ozone recovery by taking the difference between the conditions at the end of the century (2080-2099) and
current-day conditions (2011-2030). The second ensemble, FixODS, is forced only by GHGs ($CO_2$, $CH_4$ and $N_2O$) following
SSP5-8.5, while ODSs are kept fixed at their 1995 climatological annual cycle, obtained by taking the average over 1991-2000.
The majority of the ODSs peaked during this period and therefore this ensemble simulates a world in which the ozone hole
does not recover, but persists at its maximum extent. By taking the difference between INTERACT_$O_3$ and FixODS over
2075-2099 we can extract the influence exerted by ozone recovery until the end of the century. In contrast, the third ensemble,
FixGHG, is forced only by ODSs that decrease following SSP5-8.5 and GHGs are kept fixed at their 1995 climatological
annual cycle. We take the difference between INTERACT_$O_3$ and FixGHG over 2075-2099 to extract the effect that climate
change has during the twenty-first century. Finally, the fourth ensemble, PRESC_$O_3$ is identical to INTERACT_$O_3$, with the
exception that the CMIP6 ozone field (Hegglin et al., 2016) is prescribed instead of calculating the ozone interactively. The
combined effect of the increase in GHGs and ozone recovery during the twenty-first century is assessed for the case when
the CMIP6 ozone is prescribed by taking the difference between the conditions at the end of the century (2080-2099) and
current-day conditions (2011-2030) in this ensemble. This total effect is then compared to that obtained from INTERACT_$O_3$
to investigate if the treatment of ozone affects our results. Unlike in the previous CMIP phases, the CMIP6 ozone field includes
zonal variations and differs for each future GHG scenario. Therefore, the prescribed ozone field is consistent with the increase
in GHGs in SSP5-8.5, which is important as GHGs are known to affect ozone concentrations (Rosenfield et al., 2002; Waugh
et al., 2009a; Eyring et al., 2010; Revell et al., 2012). We note that the total effect in PRESC_$O_3$ should only be compared
with the total effect in INTERACT_$O_3$ and not with the individual effects of ozone recovery and increasing GHGs, because





the individual effects are likely to be different when the ozone field is prescribed. The separation of the different effects using the four ensembles described above is summarized in Table 1.

## 2.2 Residual circulation and wave diagnostics

The propagation of planetary-scale waves into the stratosphere is investigated using the three-dimensional flux of wave activity
defined by Plumb (1985) as follows:

$$
\boldsymbol{F}_s = \frac{p\,cos\phi}{p_o} \times \left\{ \begin{array}{l} \dfrac{1}{2a^2\,cos^2\phi}\left[\left(\dfrac{\partial\psi'}{\partial\lambda}\right)^2 - \psi'\dfrac{\partial^2\psi'}{\partial\lambda^2}\right] \\[2ex] \dfrac{1}{2a^2\,cos\phi}\left(\dfrac{\partial\psi'}{\partial\lambda}\dfrac{\partial\psi'}{\partial\phi} - \psi'\dfrac{\partial^2\psi'}{\partial\lambda\partial\phi}\right) \\[2ex] \dfrac{2\Omega^2\,sin^2\phi}{N^2 a\,cos\phi}\left(\dfrac{\partial\psi'}{\partial\lambda}\dfrac{\partial\psi'}{\partial z} - \psi'\dfrac{\partial^2\psi'}{\partial\lambda\partial z}\right) \end{array} \right\} \tag{1}
$$

where $\psi$ is the quasi-geostrophic streamfunction, $N^2$ is the buoyancy frequency, $a$ is the radius of the Earth, $\omega$ is the rotation rate of the Earth, $\phi$ and $\lambda$ are the latitude and the longitude respectively, p is the pressure, with $p_o$ taken as 1000 hPa, and the primes denote departures from the zonal mean.

We also examine changes in the zonal-mean residual circulation and its drivers within the Transformed Eulerian Mean (TEM; Andrews et al. 1987) framework. The Eliassen-Palm (EP) flux of wave activity, equivalent to the zonal mean of the vertical and meridional components of the Plumb flux, is defined as

$$
F_\phi = -a\cos\phi\overline{v'u'} \tag{2}
$$
$$
F_p = fa\cos\phi\frac{\overline{v'\theta'}}{\overline{\theta_p}} \tag{3}
$$

and its divergence, $(a\cos\phi)^{-1}\nabla\cdot F$, which is a measure for wave-mean flow interaction, is defined as

$$
\nabla\cdot F = \frac{1}{a\cos\phi}\frac{\partial(F_\phi\cos\phi)}{\partial\phi} + \frac{\partial F_p}{\partial p} \tag{4}
$$

$u$ and $v$ are the zonal and meridional velocity components, respectively, $f$ is the Coriolis parameter, $\theta$ is the potential temperature and $\theta_p$ is the partial derivative of $\theta$ with respect to pressure. The overbars denote the zonal mean and the primes denote departures from the zonal mean. The downward residual velocity, $\overline{w^*}$, was calculated from the residual streamfunction, $\Psi$, as

$$
\overline{w^*} = \frac{gH}{pa\cos\phi}\frac{\partial\Psi}{\partial\phi} \tag{5}
$$

where

$$
\Psi = -\frac{\cos\phi}{g}\int_p^0 \overline{v^*}(\phi,p)\mathrm{d}p \tag{6}
$$

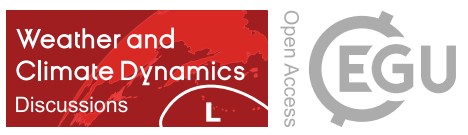

The meridional residual velocity is defined as

$$\overline{v^*} = \overline{v} - \frac{\partial}{\partial p}\left(\frac{\overline{v'\theta'}}{\overline{\theta_p}}\right) \tag{7}$$

The scale height, $H$, was taken as $7000$ m and $g$ denotes the gravitational acceleration.

## 2.3    Agulhas Leakage calculation

As the Agulhas leakage consists of rings, eddies and filaments, it is characterized by strong spatiotemporal variability and therefore cannot be properly measured using an Eulerian approach. Instead, the Agulhas Leakage is commonly quantified using Lagrangian particle tracking (Biastoch et al., 2008, 2009; van Sebille et al., 2009; Durgadoo et al., 2013; Weijer and

van Sebille, 2014; Cheng et al., 2016, 2018). Here, we follow the method of Durgadoo et al. (2013) and we use the offline ARIANE software (Blanke and Raynaud, 1997; Blanke et al., 1999) to track Lagrangian particles originating in the Agulhas Current. The virtual particles are seeded in a zonal section across the Agulhas Current at $32°$ S over the full depth for southward velocities and are advected for a maximum period of five years based on the five day three-dimensional velocity field obtained from FOCI. Each particle has a fraction of the total instantaneous Agulhas Current volume transport assigned to it that remains

constant as the particle is advected forward. An annual time series of Agulhas leakage is obtained by summing up the volume transport associated with the particles that have exited the Cape Basin through the Good Hope section, marked by the black line in Fig. 15a, within five years. This gives the fraction of the Agulhas Current volume transport that enters the Atlantic Ocean. The Agulhas leakage is computed for each ensemble member of FixODS, FixGHG, INTERACT_$O_3$ and PRESC_$O_3$ before the ensemble averaging is performed. Because of the five year-period granted for the particles to be advected, the Agulhas

leakage time series ends in 2094, although the FOCI simulations were performed until 2099. Therefore, we calculated the Agulhas leakage differences between INTERACT_$O_3$ and FixODS/FixGHG based on the 2075-2094 period and we estimate the total change in the Agulhas leakage by taking the difference between the 2080-2094 and 2014-2028 periods.

## 3    Projected ozone recovery in FOCI

Decreasing levels of ODSs and increasing levels of GHGs both lead to changes in the stratospheric ozone. Figure 1 shows the

vertical and seasonal distribution of the twenty-first century changes in Antarctic polar cap ozone due to ODSs, GHGs and due to their combined effect. Austral spring lower and middle stratospheric ozone levels exhibit a strong increase in response to declining ODSs (Fig. 1a), peaking in October, in agreement with results from other CCMs (Perlwitz et al., 2008; Son et al., 2008). This is the month during which the strongest ozone depletion occurred in FOCI in the second half of the twentieth century (Ivanciu et al., 2021). In addition, there is a significant increase in ozone throughout the whole year in the upper

stratosphere and a small decrease between $30$ hPa and $10$ hPa in December and January.

GHGs also influence the stratospheric ozone through their chemical, radiative and dynamical effects (Haigh and Pyle, 1982; Rosenfield et al., 2002; Revell et al., 2012; Chiodo et al., 2018; Morgenstern et al., 2018). In the upper stratosphere, GHGs lead to an increase in ozone of similar magnitude to the ODSs effect (Fig. 1b). At these levels, GHGs have a strong cooling



effect (Fig. 2b) owing to their emission of long wave (LW) radiation. This cooling slows down the temperature-dependent
ozone depleting reactions in the upper stratosphere, resulting in an increase in ozone (Haigh and Pyle, 1982; Jonsson et al.,
2004; Rosenfield et al., 2002; Revell et al., 2012; Morgenstern et al., 2018). In addition, chemical processes related to increases
in $CH_4$ and $N_2O$ cause an overall ozone increase and an overall ozone loss, respectively. The influences of individual GHGs
on ozone levels have been discussed in previous studies (Portmann and Solomon, 2007; Revell et al., 2012; Dhomse et al.,
2018; Morgenstern et al., 2018) and are not distinguished here. We show instead the combined influence of $CO_2$, $CH_4$ and
$N_2O$ on ozone (Fig. 1b) to illustrate that the increase in GHGs drive important changes in ozone. Therefore, the GHG-induced
dynamical changes discussed in the following sections include not only the direct effects of GHGs, driven by radiation changes,
but also indirect effects mediated through changes in ozone. The studies by Morgenstern et al. (2014) and Chiodo and Polvani
(2016) showed that these indirect effects are substantial and they offset part of the direct GHG impacts on the tropospheric
circulation.

Figure 1 shows that increasing GHGs and decreasing ODSs each contribute about half of the total ozone increase in the
upper stratosphere (Fig. 1c). This is consistent with the findings of the latest Scientific Assessment of Ozone Depletion (World
Meteorological Organization, 2018). In the lower and middle stratosphere in austral spring, the effect of declining ODSs
dominates the total ozone change. GHGs also contribute a small increase in ozone at these levels in spring (Fig. 1b), by
enhancing the residual circulation, which transports ozone-rich air to the polar cap, as discussed in Sect. 4.1.3.

In FOCI, the INTERACT_$O_3$ ensemble mean projects that the October Antarctic TCO will return to 1980s values in 2048
under the SSP5-8.5 pathway, with the spread of the individual members ranging from 2048 to 2050 (supplementary Fig. S1).
This is earlier than the CCMI ensemble mean estimate of 2060, with a $1\sigma$ uncertainty of 2055-2066 (Dhomse et al., 2018), or
the CCMI weighted mean estimate of 2056, with a 95% confidence interval of 2052-2060 (Amos et al., 2020), both of which
are obtained using RCP6.0. The discrepancy to the CMMI projected recovery date arises most likely due to higher GHGs levels
in SSP5-8.5, which accelerate ozone recovery. In the absence of future increase in GHGs (FixGHG ensemble), the October
Antarctic TCO returns to 1980 levels in FOCI only in 2067, with individual members simulating a recovery between 2066 and
2076. Our simulations therefore suggest that increasing GHGs following the SSP5-8.5 pathway accelerates Antarctic ozone
recovery by at least two decades.

Figure 1 offers a comparison between the total ozone change simulated by FOCI and that in the CMIP6 ozone field prescribed
to the PRESC_$O_3$ ensemble (panels c and d). Both fields exhibit a similar vertical and seasonal structure of ozone recovery,
with the highest increases occurring in the upper stratosphere and in spring between 100 hPa and 20 hPa. However, there are
differences in the magnitude of the ozone increase. The recovery of the ozone hole is stronger in FOCI than in the CMIP6
field. In addition, the increase in the upper stratosphere is stronger in FOCI during spring and weaker during March-June.
These differences in ozone recovery between the FOCI and CMIP6 fields contribute to the different changes in temperature
and atmospheric circulation in INTERACT_$O_3$ and PRESC_$O_3$ discussed in the following sections.





## 4 Impacts of ozone recovery and GHG increase in the Southern Hemisphere

### 4.1 Atmospheric circulation changes

#### 4.1.1 Temperature and zonal wind

Figure 2 shows the effects of ozone recovery and of increasing GHGs on the Antarctic polar cap temperature, as well as their
combined effects in the presence of interactive ozone (INTERACT_$O_3$) and of prescribed ozone (PRESC_$O_3$). A significant
warming occurs between 200 hPa and 20 hPa in austral spring and summer in response to ozone recovery (Fig. 2a), in
agreement with previous studies (Perlwitz et al., 2008; Son et al., 2009; Karpechko et al., 2010; McLandress et al., 2010). The
warming reaches its maximum in November. Above this warming, a significant cooling occurs in November and December
due to the dynamical response to ozone recovery, as discussed below. Increasing GHGs lead to a cooling of the stratosphere
throughout the year, which is strongest in the upper stratosphere (Fig. 2b), in line with the findings of McLandress et al.
(2010). We note that stratospheric temperature changes due to GHGs inferred from Fig. 2b include the warming caused by the
GHG-driven ozone increase in the middle and upper stratosphere (Fig. 1b) and that the direct radiative effect of GHGs is likely
larger. An exception to the GHG-induced stratospheric cooling occurs below 20 hPa in austral spring, when the temperature
increases as a result of increasing GHGs, but this increase is not significant when the polar cap mean is considered. We will
show below that this is caused by contrasting dynamical heating responses to GHGs over different sectors of the Southern
Ocean that compensate each other in the zonal mean. In the troposphere, there is a significant warming due to GHGs thought
the entire year, reaching 4.4 K at the surface.

The combined effect of ozone recovery and GHGs (Fig. 2c, d) is dominated by the GHG-induced cooling (warming) in the
stratosphere (troposphere) and by the ozone-induced warming in the lower stratosphere during spring. There is little difference
between the temperature changes in the ensemble with interactive ozone, INTERACT_$O_3$, and the ensemble with prescribed
ozone, PRESC_$O_3$, during the months when the GHG effect dominates. In contrast, during austral spring, when the strato-
spheric temperature changes are dictated by changes in ozone, there is clear difference between the two ensembles. PRESC_$O_3$
simulates a spring lower stratospheric maximum warming that is about half of that simulated by INTERACT_$O_3$ in the polar
cap mean. In addition, the November upper stratospheric cooling is also weaker in PRESC_$O_3$ than in INTERACT_$O_3$.

Changes in the polar cap temperature are closely related to changes in zonal wind through the thermal wind balance. Figure
3 shows the vertical profile of the zonal mean zonal wind changes averaged over November and December, when the largest
temperature signal in the lower stratosphere occurs (Fig. 2). The ozone-induced polar cap warming is associated with a weak-
ening of the stratospheric westerlies that extends to the troposphere and the surface (Fig. 3a). In contrast, the increasing GHGs
result in a positive (westerly) wind change extending from the top of the stratosphere to the surface in the mid-latitudes (Fig.
3b). The GHG effect peaks at the top of the tropospheric westerly jet around 25° S and occurs throughout the year, unlike the
ozone effect, which occurs only during spring in the stratosphere and summer in the troposphere. The combined ozone and
GHG effect consists of a weakening of the stratospheric westerlies south of 60°S, where the ozone effect dominates, and a
strengthening at the top of the tropospheric jet and on its poleward flank, where the GHG effect dominates (Fig. 3c, d). The





stratospheric westerlies weakening has a much lower magnitude in PRESC_$O_3$ than in INTERACT_$O_3$, in agreement with the
350  lower magnitude of the total spring temperature change. At the same time, the positive change in the midlatitudes is stronger
and exhibits a larger latitudinal extent in PRESC_$O_3$ both in the stratosphere and in the troposphere, as the GHG westerly
effect is offset to a lesser degree by the weaker ozone easterly effect in this ensemble.

The Antarctic ozone hole is not centered over the polar cap (e.g., Grytsai et al., 2007) and therefore the ozone recovery also
occurs displaced from the South Pole. In addition, the dynamical effects of both ozone recovery and increasing GHGs exhibit
355  strong zonal variations. Therefore, it is interesting to investigate the horizontal structure of the stratospheric changes. Figures
4 and 5 show maps of the October ozone and temperature changes at 50 hPa and of the zonal wind changes at 20 hPa. The
region of large ozone recovery in response to declining ODS is slightly displaced towards the Atlantic sector (Fig. 4a), causing
the warming response to also extend towards the Atlantic (Fig. 4b). This displacement coincides with the displacement of the
ozone hole simulated by FOCI for the latter part of the twentieth century (Ivanciu et al., 2021). There is a significant ozone
increase due to GHGs focused over the Pacific sector (Fig. 4d), which occurs due changes in dynamics, as discussed below.
This agrees to the zonally asymmetric ozone response to a quadrupling of $CO_2$ reported from four CCMs by Chiodo et al.
(2018), characterized by a positive change over the Pacific and a negative one over the Indian Ocean.

Even more interesting is the temperature response to increasing GHG (Fig. 4e), which is characterized by a dipole, with
cooling occurring towards South Africa and warming occurring towards the Pacific, in the same region as the strongest ozone
increase. We note that the magnitude of the GHG-induced October warming reaches 7 K, only 1 K lower than the maximum
warming associated with ozone recovery of 8 K, while the maximum GHG-induced cooling reaches -6 K. This indicates
that GHGs bring an important contribution to the total spring temperature changes at this height. This contribution is largely
underestimated if zonally-averaged fields are investigated (Fig. 2b), as the cooling pole offsets a large part of the warming
pole. The total October lower stratospheric temperature change (Fig. 5b) illustrates the combination of the ozone and GHG
effects well. The ozone-induced warming persists over Antarctica, but it is extended towards the Pacific by the superposition of
the GHG-induced warming. The cooling due to GHGs dominates over the Southern Ocean towards South Africa and at lower
latitudes.

The dipole structure of the spring temperature response to GHGs is associated with zonal wind changes of opposite sign in
the Eastern and Western Hemispheres (Fig. 4f). The Pacific sector is characterized by a weakening of the polar night jet on its
equatorward flank and a strengthening on its poleward flank, while the opposite occurs over the Atlantic and Indian sectors.
This implies a shift of the polar vortex towards South Africa in austral spring driven by increasing GHGs. In contrast, ozone
recovery leads to a circumpolar weakening of the polar night jet (Fig. 4c). The total stratospheric zonal wind change in October
(Fig. 5c) consists of a weakening of the polar night jet, which peaks over the Pacific sector, where the GHG effect reinforces
the weakening due to ozone recovery. The maximum weakening in this region reaches 16.5 m s$^{-1}$ or 41% of the current
day October climatological strength of the polar vortex. At midlatitudes over the Atlantic and Indian sectors the GHG effect
dominates and the outer flank of the polar vortex strengthens, while the easterlies weaken. The maps showing the temperature
and zonal wind changes in Figs. 4 and 5 demonstrate that important GHG-induced changes are missed if only changes in zonal
mean fields are investigated.





The total zonal wind change in the PRESC_$O_3$ ensemble (Fig. 5f) resembles that caused by GHGs, indicating that the ozone

effect is weaker in this ensemble. The magnitude of the polar vortex weakening is significantly larger in INTERACT_$O_3$ than in PRESC_$O_3$ over the Pacific sector (Fig. 5i). At the same time, the strengthening over the Indian sector is weaker in INTERACT_$O_3$ than in PRESC_$O_3$. These differences arise from the different temperature response in the two ensembles (Fig. 5b, e and h). The total temperature change in PRESC_$O_3$ retains more of the dipole structure caused by GHGs. This suggests that the increase in GHGs plays a more dominant role in driving the temperature changes in this ensemble. The region

of increasing temperature is shifted eastward and the maximum warming is weaker by about 3 K in PRESC_$O_3$ compared to INTERACT_$O_3$. These differences in the temperature response can be partly explained by the patterns of ozone recovery in INTERACT_$O_3$ and prescribed to PRESC_$O_3$ (Fig. 5a, d). There is a stronger increase in ozone over the Pacific sector in INTERACT_$O_3$ (Fig. 5g), with ozone recovery peaking between the Antarctic Peninsula and the Ross Sea this ensemble (Fig. 5a). In contrast, ozone recovery peaks over the Weddell Sea in the CMIP6 field (Fig. 5b). The different ozone recovery

patterns cause the temperature response to ozone recovery to offset and reinforce the response to increasing GHGs in different regions in the two ensembles. At the same time, the smaller ozone increase in PRESC_$O_3$ leads to an overall weaker response to ozone recovery and to GHGs playing a more important role in this ensemble. We also note that the shape and the magnitude of the current day ozone hole differs between INTERACT_$O_3$ and PRESC_$O_3$ (contours in Fig. 5a and d), with the ozone hole in PRESC_$O_3$ being displaced towards South Africa and the ozone hole in INTERACT_$O_3$ being displaced towards South

America.

The temperature changes discussed above arise from changes in the radiative and dynamical heating rates. Figures 6 and 7 illustrate the heating rate changes in response to ozone recovery and increasing GHGs individually and combined, respectively. The increase in ozone as ODS are eliminated from the stratosphere results in more SW radiation being absorbed (Fig. 6a). Therefore, the effect of ozone recovery dominates the total change in the SW heating rate (Fig. 7a), but there is also a small

contribution arising from the GHG-driven spring increase in ozone (Fig. 6d). The total change in the SW heating rate (Fig. 7a, d) illustrates the effect of the differences in ozone recovery on the temperature change in INTERACT_$O_3$ and PRESC_$O_3$. In agreement with the stronger ozone increase in INTERACT_$O_3$, there is a stronger SW warming over Antarctica and extending to the South Pacific in this ensemble compared to PRESC_$O_3$ (Fig. 7a, d and g). This contributes to the larger temperature increase simulated by INTERACT_$O_3$ (Fig. 5b). Interestingly, the region over the Pacific sector where ozone levels (Fig. 5g)

and SW heating (Fig. 7g) are higher in INTERACT_$O_3$ compared to PRESC_$O_3$ coincides with region of GHG-induced ozone increase (Fig. 4d). The effect of GHGs on ozone represents a dynamical feedback that cannot be simulated in PRESC_$O_3$ because the ozone field is prescribed and therefore cannot react to changes in dynamics. Although this GHG effect is captured in the two CCMs that produced the CMIP6 ozone field prescribed in PRESC_$O_3$, and although this ozone field is consistent with the SSP5-8.5 scenario used here, intermodel differences in simulating the ozone response to increasing GHGs and declining

ODSs lead to inconsistencies between ozone and dynamical changes when the ozone field obtained from other models is prescribed. This is emphasized by the weak ozone and SW heating rate changes in PRESC_$O_3$ in the region where the dynamical increase in ozone driven by GHGs occurs in FOCI.



There are also important changes in the dynamical heating rate (Fig. 7c, f), of similar magnitude to the SW heating rate changes. The Drake Passage and the eastern half of the Pacific sector exhibit pronounced dynamical warming, while East Antarctica and the Indian Ocean sector exhibit a strong cooling. This pattern is driven by the dynamical response to increasing GHGs (Fig. 6f). The dynamical heating rate change due to GHGs explains the temperature response to GHGs almost entirely, with an additional weak contribution due to the increase in the SW heating rate over the Pacific sector caused by the dynamical ozone increase in this region. The change in the dynamical heating rate due to ozone recovery is weak in October at 50 hPa (Fig. 6c), but it becomes stronger in November, acting to cool the polar cap (Fig. S2l). The region of dynamical warming is larger and the dynamical cooling is stronger in PRESC_$O_3$ (Fig. 7f) than in INTERACT_$O_3$ (Fig. 7c). These differences in the dynamical heating rate originate from differences in wave activity and in the residual circulation, which are discussed in the following two sections. The LW heating rate changes act to dampen the warming due to SW and dynamical heating changes in all simulation (Figs. 6b, e and 7b, e).

In summary, ozone recovery during the twenty-first century reverses the effects of ozone depletion. It leads to a springtime warming of the polar cap in the lower stratosphere and an associated weakening of the westerly winds that extends to the surface in austral summer. In contrast, increasing GHGs lead to a cooling of the stratosphere throughout most of the year and an acceleration of the subtropical jet that extends to the top of the stratosphere. A significant exception to the otherwise rather zonally uniform changes occurs in austral spring, when a dynamical response to increasing GHGs results in contrasting temperature and zonal wind changes over the Pacific and Indo-Atlantic sectors that offset each other in the zonal mean and are not detectable if only zonal mean fields are investigated. The importance of these GHG-driven changes, which are reported here for the first time, is emphasized by their magnitude, which is similar to the magnitude of the changes due to ozone recovery. In the months and regions where the ozone effect is important, the combined ozone and GHG effect is stronger in the stratosphere in the ensemble with interactive ozone chemistry than in the ensemble with prescribed ozone, as the ozone effect is stronger in the former ensemble. This demonstrates the importance of the ozone field in setting the magnitude of the temperature and zonal wind changes during the twenty-first century.

### 4.1.2 Wave activity

The flux of wave activity from the troposphere into the stratosphere peaks in the SH during spring, disturbing the polar vortex and driving downwelling and dynamical heating over the high latitudes. The direction of wave activity propagation in three dimensional space can be investigated using the Plumb flux (Plumb, 1985). When zonal averages are considered, the vertical and meridional components of the Plumb flux reduce to the EP flux, which will be discussed in the next section. Figure 8 shows maps of changes in the Plumb flux at 50 hPa during October and November. Additional maps for the upper stratosphere can be found in the supplement (Figs. S2 and S3).

The increase in GHGs drives strong changes in wave activity during the twenty-first century. During austral spring, it enhances the flux of wave activity over the Indian Ocean sector and diminishes it over the Pacific and Atlantic sectors (Fig. 8b, f). These changes persist throughout the depth of the stratosphere and indicate that increasing GHGs enable planetary waves to propagate higher into the stratosphere and mesosphere over the Indian Ocean during spring. This results in reduced (enhanced)



wave breaking in the middle (upper) stratosphere, which in turn drives a weakening (strengthening) of the downwelling in the middle (upper) stratosphere. The dynamical cooling (warming) at 50 hPa (5 hPa) seen in Fig. 6f (Fig. S2m) in this region occurs as a consequence of the weakened (strengthened) downwelling. Over the Pacific sector, the propagation of wave activity

is inhibited by increasing GHGs. Consequently, planetary waves break at lower levels and the wave drag is enhanced in the middle stratosphere and weakened above, driving stronger downwelling in the lower to middle stratosphere and weaker downwelling in upper stratosphere. As a result, the lower and middle stratosphere warms over the Pacific sector (Fig. 6f), while the upper stratosphere cools (Fig. S2m). The GHG-induced changes in the flux of wave activity are closely related to the changes experienced by the polar vortex (Fig. 4f). The acceleration of the polar night jet on its outer flank over the Indian sector shifts

the location of the zero zonal wind velocity towards the equator. The region of zero zonal wind velocity represents the critical line for Rossby wave propagation (Dickinson, 1968) and therefore this shift leads to increased upward wave propagation on the equatorward side of the polar vortex. The opposite occurs over the Pacific sector, where the weakening of the polar night jet on its outer flank shifts the polar vortex poleward and reduces the ability of planetary waves to propagate upward. The mechanism at work in both sectors is essentially the same as the "critical layer control" of wave breaking discussed by Shepherd

and McLandress (2011) for the subtropical lower stratosphere, whereby a strengthening of the upper flank of the subtropical jet shifts the critical layers upward, resulting in more wave activity entering the stratosphere. We note that the GHG-induced changes in the stratospheric winds and the associated changes in the flux of planetary wave activity exhibit a strong seasonality. The change towards more easterly velocities occurs only between July and November, peaking in September and October. During the rest of the year, the change towards more westerly velocities occurs circumpolarly in the SH.

In contrast to the effect of increasing GHGs, the effect of ozone recovery on wave activity is weak in October at 50 hPa (Fig. 8a), in agreement with the weak dynamical heating changes in Fig. 6c. There are, however important ozone-related changes in the upper stratosphere, where there is a weakening of the flux of wave activity in response to ozone recovery, particularly over the Atlantic sector and Drake Passage. The reduction in the planetary wave forcing leads to weaker downwelling over the polar cap and dynamical cooling (Fig. S2i). This effect becomes more pronounced, circumpolar and it extends into the

lower stratosphere in November (Figs. 8e and S2l), offsetting part of the radiative warming effect of ozone recovery. As was the case for the GHG effect, the changes in the wave activity flux due to ozone recovery are linked to changes in the polar night jet. As the jet is decelerated in October in response to the ozone-induced polar cap warming (Fig. 4), the critical line for wave propagation migrates poleward, inhibiting wave propagation. Since the polar vortex in FOCI is displaced towards South America (contours in Fig. 5c), the inhibition of wave propagation is initiated in this region (Fig. S2a). The effect then propagates

downward together with the weakening polar night jet in November and December, and extends around the Antarctic continent (Figs. 8e and S2c) as the breakdown of the polar vortex is accelerated by ozone recovery.

When the effects of ozone recovery and increasing GHGs are considered together, the GHGs effect dominates the October wave activity changes and can be clearly discerned in both the ensemble with interactive ozone, INTERACT_$O_3$ and the ensemble with prescribed ozone, PRESC_$O_3$, although some differences in the magnitude of the changes can be distinguished

(Figs. 8c, d and S3a ,b). In particular, the strengthening of the wave activity flux is more pronounced in INTERACT_$O_3$ than



in PRESC_O$_3$ at 50 hPa (Fig. 8c, d), while the weakening of the flux is more pronounced in PRESC_O$_3$ at 5 hPa (Fig. S3a, b).

In November, the intensification of the wave activity flux over the Indian Ocean due to GHGs decreases and the weakening over the Pacific becomes more pronounce at 50 hPa (Fig. 8f). Over the Indian sector, the GHG and ozone effects offset each
other in the upper stratosphere (Fig. S3c), while the ozone effect dominates in the lower stratosphere (Fig. 8g). Interestingly, the GHG effect still dominates in this region in PRESC_O$_3$ in the upper stratosphere (Fig. S3d), implying that the ozone effect is weaker in this ensemble. Over the Atlantic and Pacific sectors at 50 hPa and over the Pacific alone at 5 hPa, the effects of ozone recovery and increasing GHGs reinforce each other, resulting in a stronger weakening of the wave activity flux (Figs. 8g and S3c). All these changes in the flux of wave activity lead to changes in the residual circulation, which we explore further in
the next section.

In summary, during austral spring, the increase in GHGs enables planetary waves to propagate higher into the stratosphere over the Eastern Hemisphere and inhibits wave activity propagation over the Western Hemisphere. Consequently, wave breaking is reduced (enhanced) in the middle (upper) stratosphere, driving a weakening (strengthening) of the polar downwelling and dynamical cooling (warming) over the Eastern Hemisphere, while the opposite occurs over the Western Hemisphere. Ozone
recovery leads to a weakening of the flux of wave activity, which is initiated in the upper stratosphere in October and extends to the lower stratosphere in November, driving a weakening of the downwelling in the lower stratosphere and dynamical cooling that partially offsets the radiative warming effect of ozone recovery during this month. The changes in wave activity are sensitive to the ozone field that is used, with both the magnitude and the pattern of change differing between the ensembles with interactive and with prescribed ozone.

### 4.1.3 The Brewer-Dobson circulation

Although the BDC comprises both the mean meridional residual circulation and two-way mixing processes, here we only investigate the contributions of ozone recovery and increasing GHGs to the twenty-first century changes in the residual circulation. As a first measure of the changes in the strength of the BDC, we analyze the tropical upward mass flux and the extratropical downward mass flux in each hemisphere, shown in Fig. 9 at the 50 hPa level for each month. In both hemispheres, the impact
of increasing GHGs is a clear strengthening of the downward mass flux and hence of the BDC, peaking between mid-fall and mid-spring and ceasing in late spring and early summer. For the Northern Hemisphere (NH), the peak GHG-induced strengthening of the BDC in winter agrees with the results of previous studies (e.g., Shepherd and McLandress, 2011; Chrysanthou et al., 2020). For the SH, our results are at odds with those of Chrysanthou et al. (2020) who reported a peak strengthening of the BDC in austral summer in response to a quadrupling of CO$_2$ in an atmospheric model with prescribed preindustrial
ozone. We note that the GHG-induced strengthening of the residual mean streamfunction is confined south of 55°S in austral winter, unlike in summer when it reaches the polar region. The magnitude of the strengthening during winter is, however, much larger than that during summer. An important exception to the GHG-induced strengthening occurs in November in the SH, when GHGs lead to a weakening of the mass flux in the lower stratosphere. The upward tropical mass flux is enhanced by increasing GHGs throughout the year, driven from the NH between November and February in agreement with the simulations





by Oberländer et al. (2013), from the SH between May and July, and from both hemispheres to different extents for the rest of the year. In contrast, ozone recovery weakens the BDC, which is consistent with previous studies (Oman et al., 2009; Lin and Fu, 2013; Oberländer et al., 2013; Polvani et al., 2018, 2019). The ozone effect is strong in the lower stratosphere only in November-December in the SH and April-May in the NH. The largest impact occurs in the SH in November, when the recovery of the Antarctic ozone hole takes place. This is also the only month when the tropical mass flux is significantly affected by

ozone recovery.

    Throughout most of the year, the total change in the BDC during the twenty-first century follows the change due to increasing GHGs and the BDC strengthens. In November and December, however, the BDC in the SH weakens, driven by both ozone recovery and increasing GHGs in November and by ozone recovery alone in December. The weakening in the SH offsets the strengthening in the NH during November, leading to an insignificant change in the tropical mass flux. The change in the

residual mean stream function shown for November in supplementary Fig. S4 reveals that while ozone recovery weakens the BDC throughout the depth of the stratosphere, the increase in GHGs only weakens the lower branch.

    The two months during which ozone recovery influences the BDC are also the two months during which all three members of the INTERACT_$O_3$ ensemble simulate a stronger weakening than all three members of the PRESC_$O_3$ ensemble (hallow circles in Fig. 9b). This once again points to the fact that the effects of ozone recovery are weaker when the CMIP6 ozone field

is prescribed. A similar behavior occurs in May in the NH, although the ozone recovery has a weaker effect there. In addition, the November BDC weakening is confined below 20 hPa in PRESC_$O_3$, exhibiting a structure more similar to that of the GHG effect, while the weakening in INTERACT_$O_3$ extends to the upper stratosphere, in accordance with the ozone effect (Fig. S4).

    In order to link the changes in wave activity due to GHGs and ozone recovery discussed in Sect. 4.1.2 to the changes in the

BDC presented above, we show in Fig. 10 the changes in the polar cap vertical residual velocity for each calendar month. In agreement with the ozone-induced weakening of the mass flux (Fig. 9a) and the dynamical cooling of the polar cap (Fig. S2k, l), the downwelling over the polar cap is considerably reduced in November throughout the stratosphere due to ozone recovery (Fig. 10a). This reduction is accompanied by an anomalous divergence of the EP flux (Fig. S6a) and a positive change in the eddy heat flux (Fig. S5a), indicating reduced wave activity propagation. The zonal mean eddy heat flux is equivalent to the

zonal mean of the vertical component of the Plumb flux shown in Fig. 8, with negative eddy heat flux values indicating upward wave activity propagation. The positive eddy heat flux change due to ozone recovery therefore agrees with the negative change in the vertical component of the Plumb flux shown in Figs. 8 and S2. The decrease in polar downwelling due to ozone recovery begins in October in the upper stratosphere, propagates downward and persists in the lower stratosphere until February, but with a much lower magnitude. It is related to the reduction of the polar vortex strength due to ozone recovery (Figs. 3a and

4c), which implies an earlier break-down of the polar vortex in November and a decrease in the height of the critical line for Rossby wave propagation during summer. These changes inhibit the upward propagation of wave activity and, as wave activity cannot reach as high as before, lead to reduced wave breaking (the anomalous EP flux divergence seen in Fig. S6a) originating in the upper stratosphere and extending downward with time. The weakening of the BDC in response to ozone recovery is therefore explained by the changes in wave breaking. The good agreement between the zonal mean fields presented here and





the three-dimensional Plumb flux attests that the zonal mean fields correctly capture the changes in wave activity and the BDC caused by ozone recovery.

We additionally note that ozone recovery leads to anomalous convergence between 50 and 7 hPa in October (Fig. S6a) and a weak but significant strengthening of the lower stratospheric downwelling (Fig. 10a). This corroborates the results of McLandress et al. (2010), who found an increase in the polar cap downwelling in the lower stratosphere in spring driven by

enhanced EP flux convergence due to ozone recovery, changing sign in summer. Furthermore, these changes during October bring confidence in the mechanism inferred from the changes in the Plumb flux, whereby wave propagation is inhibited in the upper stratosphere (Fig. S2a) and wave breaking occurs at lower levels, leading to a reduction of the downwelling in the upper stratosphere and dynamical cooling (Fig. S2i).

The increase in GHGs also exerts the largest influence on the SH polar cap downwelling in austral spring (Fig. 10b). The

relatively small impact of GHGs on the polar downwelling during the rest of the year might seem at odds with the peak GHG-induced increase in the downward mass flux between April and October (Fig. 9a). This discrepancy is explained by seasonal differences in the latitude range at which the BDC is affected by the GHG increase. During winter, although the GHG effect on the BDC has the largest magnitude, it is restricted to the low and midlatitudes, as the propagation of wave activity is enhanced towards the equator (not shown). During spring, the GHG effect extends to the polar latitudes affecting the polar downwelling,

although its overall magnitude is weaker. McLandress et al. (2010) also showed that important changes in the SH downward mass flux caused by increasing GHGs occur outside of the polar cap. Therefore, the seasonality of the polar downwelling and of the SH total downward mass flux magnitudes are not directly comparable. The effects of ozone recovery on the polar cap downwelling and on the total SH downward mass flux are in better agreement, as the ozone-induced BDC changes are driven from the polar region.

Figure 10b shows that GHGs strengthen the downwelling above 20 hPa in October. As explained in Sect. 4.1.2, a strengthening of the downwelling in the upper stratosphere corresponds to the enhanced flux of wave activity seen over the Indian Ocean in Fig. 8b and the consequent increase of the height at which waves break. This indicates that the changes over this sector dominate in the zonal mean, as expected from their larger magnitude compared to the changes that occur over the Pacific sector. In contrast, during November the GHG effect over the Indian sector weakens and the effect over the Pacific strengthens

and dominates in the zonal mean (Fig. 8f). This can also be seen by the positive change in the eddy heat flux during this month (Fig. S5b), which implies weaker upward wave propagation. Wave breaking is reduced throughout the middle stratosphere (Fig. S6b) and, as a result, the polar downwelling weakens. In fact, as shown in supplementary Fig. S4b, the entire lower branch of the BDC in the SH weakens in November, in agreement with the reduction in the downward mass flux.

The changes in the zonal mean fields due to the increase in GHGs discussed above only partly characterize the GHG effect

on the BDC. In contrast to the changes caused by ozone recovery, the response of the SH BDC to increasing GHGs, as inferred from changes in the three-dimensional flux of wave activity, exhibits strong zonal variations that are most pronounced in austral spring. Zonal averaging illustrates only the changes occurring in the dominating sector, at best. At worst, it can tell a false story of insignificant GHG-induced changes in the BDC when the contrasting effects occurring in different regions compensate each other. This occurs, for example, in the lower stratosphere during September and October (Fig. 10b).





Similarly, the combined effect of ozone recovery and increasing GHGs on the BDC shown in Figs. 10 and S4-S6 reveals only limited information about the features of the BDC change. While it correctly captures the November weakening of the polar cap downwelling and of the BDC driven by both ozone and GHGs increases, it misses the fact that the two effect reinforce each other only over certain regions and compensate each other over other regions. Furthermore, different levels of compensation in INTERACT_O$_3$ and PRESC_O$_3$ lead to discrepancies in the timing and strength of the weakening in downwelling (Fig.

10c, d). It is possible that the degree of compensation or reinforcement of the two effects in different models add a source of uncertainty in the projected BDC changes.

It has been demonstrated in previous studies that an important part of the effect of increasing GHGs on the BDC is mediated through the rise in sea surface temperature (SST), which causes a strong warming of the tropical troposphere that accelerates the subtropical jets, resulting in a stronger BDC in the lower and middle stratosphere (Oman et al., 2009; Oberländer et al.,

2013; Chrysanthou et al., 2020). Our use of a chemistry climate model with a fully interactive ocean insures that changes in SSTs and in the atmospheric circulation can influence each other and therefore evolve in a consistent manner. Furthermore, the enhanced resolution in the Tropical Atlantic (among other regions) achieved via the INALT10X nest considerably alleviates the Tropical Atlantic warm bias present in current climate models, as demonstrated by Matthes et al. (2020). Given the particular importance of tropical SSTs in mediating the BDC response to increased GHGs (Oman et al., 2009; Oberländer et al., 2013),

simulating the Tropical Atlantic SSTs in a more realistic manner should improve the response of the BDC to the GHG forcing.

In summary, the BDC in the SH strengthens throughout most of the year driven by the increase in GHGs, but weakens in November and December, driven by both ozone recovery and increasing GHGs in November and by ozone recovery alone in December. The springtime GHG effect on the BDC exhibits strong zonal variations and the zonal mean BDC change does not properly characterize this effect, being set by the changes in the Eastern Hemisphere in October and by the changes

in the Western Hemisphere in November. During the months in which ozone recovery significantly affects the BDC, the magnitude of the change in the ensemble with interactive ozone is significantly larger than in the ensemble with prescribed ozone. Furthermore, while the changes in the former ensemble occur throughout the depth of the stratosphere, the changes in the latter ensemble are only significant up to the middle stratosphere. This shows that the effect of ozone recovery on the BDC is weaker when the CMIP6 ozone field is prescribed.

**4.1.4  Surface impacts**

We now turn our attention to the projected twenty-first century changes in the surface circulation. As shown in Sect. 4.1.1, the weakening of the stratospheric westerlies due to ozone recovery extends to the surface in November and the austral summer. Figure 11 shows maps of annual changes in the mean sea level pressure (SLP), surface zonal wind and the associated wind stress curl, in response to ozone recovery and increasing GHGs, separately. We chose to examine the changes occurring over

the entire year rather than seasonal subsets, as they are the most relevant for the long-term trends in the ocean circulation. For completion, we show additional maps of zonal wind changes during November-January and June-August in supplementary Fig. S7. Although occurring during a single season, the ozone effect is still significant in the annual mean and represents a weakening of about 4% of the current-day annual mean westerlies in the latitude band between 45°S and 60°S. In the



November-January average, the weakening is larger, reaching 8%. The change towards more easterly velocities occurs on the
poleward flank of the climatological westerlies (contours in Fig. 12b) and implies an equatorward shift in their position. As
expected, ozone recovery reverses the strengthening and poleward shift of the westerlies caused in the past by the formation
of the Antarctic ozone hole, consistent with previous findings (Perlwitz et al., 2008; McLandress et al., 2011). This can also
be seen in the effect of ozone recovery on SLP (Fig. 11a), which exhibits a strong increase over the Antarctic continent and
decreases at midlatitudes, signaling that the SAM shifts more towards its negative phase. This is confirmed by the negative
change in the SAM index shown in Fig. 13a.

Increasing GHGs, in contrast, continue the past trend towards a more positive SAM (Figs. 11d and 13a) and strengthened
surface westerlies that shift poleward (Figs. 11e and 13b, c). The magnitude of the GHG effect is much larger than that of
the ozone effect on the westerlies in the annual mean, leading to a strengthening of 17% in the mean over the latitude band
between 45°S and 60°S. As a consequence, when both effects are considered together (Fig. 12a), the GHG effect dominates
and the westerlies continue to strengthen and shift poleward during the twenty-first century (Fig. 13b, c), accompanied by a
shift toward the positive phase of the SAM (Figs. 12a and 13a). There are marked differences in the magnitude of the total
SAM and westerlies change between INTERACT_$O_3$ and PRESC_$O_3$ (Figs. 12 and 13). Significant difference between the
two ensembles occur particularly over the Indian and Pacific Oceans for both zonal wind and SLP and additionally over the
entire Antarctic continent for SLP, with changes of both sign being stronger in PRESC_$O_3$ (Fig. 12). Averaged over 45°S
and 60°S, the westerlies strengthen by 9% compared to the current day in INTERACT_$O_3$, while they strengthen by 16% in
PRESC_$O_3$. Given that the difference between INTERACT_$O_3$ and PRESC_$O_3$ lies in the treatment of ozone chemistry, the
stronger changes in PRESC_$O_3$ in the direction driven by GHGs shows that the ozone effect is much weaker in PRESC_$O_3$.
This is consistent with the results presented in the previous sections for the stratosphere, where the impact of ozone recovery
is also weaker in this ensemble.

We stress that the dominance of the GHG effect over the effect of ozone recovery on the surface winds is strongly depended
on the scenario used for GHGs. Here, we use SSP5-8.5, which is characterized by high GHG emissions throughout the twenty-
first century. CMIP6 models project that, for the "sustainability" scenario SSP1-2.6, the ozone effect dominates, leading to an
overall weakening of the westerly jet, while for the "middle of the road" scenario SSP2-4.5 the two effects compensate each
other (Bracegirdle et al., 2020). Our results are consistent with CMIP6 projections for SSP5-8.5 (Bracegirdle et al., 2020).
Although the GHG effect dominates if GHG emissions remain high during the twenty-first century, it important to note the
significance of the Montreal Protocol in reducing the impacts of climate change, as the changes in the surface westerlies are
stronger in the absence of ozone recovery.

In summary, ozone recovery drives a significant weakening of the surface westerlies and a change towards a more negative
SAM, reversing the trends caused by ozone depletion, while increasing GHGs drive a continued strengthening of the westerlies
and a change towards a more positive SAM. The GHG effect dominates in the annual mean under SSP5-8.5, but there are
large differences in the magnitude of the total effect between the ensemble with interactive ozone and the ensemble with
prescribed ozone. The effect of the ozone recovery on the surface westerlies and the SAM is weaker when the CMIP6 ozone
field is prescribed, leading to a greater dominance of the GHG effect and consequently to changes of greater magnitude. The





magnitude of the differences between the ensembles with interactive and with prescribed ozone is at least as large as the effect
of ozone recovery itself.

## 4.2   Ocean circulation changes

### 4.2.1   Agulhas Leakage

The westerly wind changes induced in the past by ozone depletion led to an increase in the inter-ocean exchange of warm and
saline waters around South Africa, known as the Agulhas leakage (e.g., Biastoch et al., 2009). In this section, we investigate
if the Agulhas leakage will change during the twenty-first century in response to ozone recovery, increasing GHGs and their
combined effects. Figure 13f and g shows the change in the Agulhas leakage and the Agulhas Current at the end of the century.
The current-day (2014-2028) mean Agulhas leakage in the ensemble with interactive ozone is 11.6 Sv (1 Sv = $10^6$ m$^3$ s$^{-1}$),
lower than the new observational estimate of 21.3 $\pm$ 4.7 Sv (Daher et al., 2020) or the older, widely-used estimate of 15 Sv
(Richardson, 2007), but comparable with the mean values of 11.2 Sv and 12.2 Sv simulated by the high resolution coupled
climate model employed by Cheng et al. (2016) and Cheng et al. (2018), respectively. For comparison, the mean Agulhas
leakage simulated by the ocean component of FOCI, using the same high resolution nest INALT10X, is 12.2 Sv (Schwarzkopf
et al., 2019). The current-day Agulhas Current transport in the FOCI ensemble with interactive ozone is 82.8 Sv, falling within
the uncertainty range of the observational estimate of 77 $\pm$ 32 Sv (Beal and Elipot, 2016). The Agulhas Current transport in
the FOCI is slightly larger than the 76.3 Sv reported by Schwarzkopf et al. (2019) from an uncoupled ocean simulation with
the INALT10X nest driven by relative wind stress. Although the ocean-atmosphere coupling in FOCI implies that the effect of
the ocean circulation on the surface stress is simulated, the much lower resolution of the atmosphere compared to the ocean
results in a smoothening of this effect, explaining why the coupled model simulates a larger Agulhas Current transport. Note
that, as explained in Sect. 2.3, the calculation of the Agulhas leakage results in a time series ending in 2094 and we adjusted
the periods over which perform our analysis accordingly (see Sect. 2.3 for details). Ozone recovery drives an Agulhas leakage
decrease of 0.9 Sv or about 7% of the current-day mean transport. This opposes the effect of increasing GHGs, which drive
an increase in leakage of 3.3 Sv or 28%. The total effect in the ensemble with interactive ozone chemistry, INTERACT_O$_3$
is an increase in leakage of 1.5 Sv or 13%, significantly lower than the increase which would occur from increasing GHGs
alone. This highlights the importance of ozone recovery in mitigating some of the ocean circulation changes resulting from
increasing GHG during the twenty-first century. In contrast, in the ensemble with prescribed ozone, PRESC_O$_3$, in which the
ozone effect is weaker, the Agulhas leakage increase by 3.4 Sv or 33%, an increase of similar magnitude to that resulting from
the GHG forcing alone when ozone is calculated interactively. These results are consistent with the changes in the westerly
winds presented in Sect. 4.1.4 and underline that the choice of how to account for stratospheric ozone changes affects the
prediction of not only the tropospheric, but also the oceanic circulation changes. This is further confirmed by the predicted
changes in the transport of the Agulhas Current at 32°S (Fig. 13g). Ozone recovery drives an increase in transport of 1.7 Sv or
2%, while increasing GHGs drive a decrease of 10.9 Sv or 13% of the current-day mean Agulhas Current transport of 82.8 Sv,





resulting in a total decrease of 7.3 Sv or 9% in INTERACT_O$_3$. In contrast, the decrease in PRESC_O$_3$ is 14.8 Sv or 17%, significantly greater than that occurring in INTERACT_O$_3$.

It might seem contrary to expectations that the Agulhas leakage increases when the transport of the Agulhas Current de-
creases. While Rouault et al. (2009) linked the past increase in the Agulhas leakage to an increase in the Agulhas Current transport driven by increased wind stress curl over the Indian Ocean associated with strengthened trade winds, van Sebille et al. (2009) found that the increase in leakage is associated with a decrease in the Agulhas Current transport. In a more recent study, Loveday et al. (2014) used a suite of models with different resolutions and performed perturbation experiments in which the strength of the trade winds was altered between the equator and 32.5°S, with no changes imposed at the latitudes of the
Agulhas leakage. They found that although the Agulhas Current transport increased almost linearly with the wind stress in all models, the response of the Agulhas leakage diverged between the low resolution models, while the high resolution mod-els, which are capable of resolving mesoscale processes and simulate a realistic leakage, did not show a significant change in leakage. Therefore, the changes in the Agulhas leakage and Agulhas Current transport appear to be decoupled. While the Agulhas Current changes seem to be driven by the trade winds, the changes in leakage are instead driven by the westerlies
(e.g., Durgadoo et al., 2013). In our simulations, there is a weakening of the westerlies on their equatorward edge and in the transition region between westerlies and easterlies driven by increasing GHGs (Figs. 11e and 12b, e), but no change in the trade winds farther north. There is, however, a good correspondence between the Agulhas Current transport at 32°S (Fig. 13g) and zonal wind anomalies over the Indian Ocean at the same latitude (Fig. 13h), with ozone-induced easterly wind anomalies associated with an increase in transport and GHG-induced westerly anomalies associated with a decrease in transport. As a result of the zonal wind changes, a decrease in the positive wind stress curl due to increasing GHGs can be seen at this latitude
in Figs. 11f and 12f, likely driving the decrease in the Agulhas Current transport.

The mechanism resulting in increased Agulhas leakage has been explored by Durgadoo et al. (2013) in perturbation exper-iments in which either the intensity or the position of the SH westerlies south of 35°S were varied. Durgadoo et al. (2013) found that strengthening the westerlies leads to more Agulhas leakage, while a poleward shift in the position of the westerlies leads to less Agulhas leakage. They explained the response of the leakage to stronger westerlies through positive wind stress
curl anomalies at the latitudes of the westerly winds, which strengthen the positive wind stress curl over the southern part of the Indian Ocean, north of the maximum westerlies. The more positive wind stress curl drives an intensified Ekman pumping and northward Sverdrup transport across the southern part of the subtropical supergyre linking the South Atlantic and Indian Oceans, and the Agulhas leakage has to intensify to provide the westward mass transport required by continuity. The changes in wind stress curl simulated by FOCI for the twenty-first century (Figs. 11c, f, 12c, f and 13e) seem to support this mecha-
nism. The GHG-induced westerlies changes lead to positive wind stress curl anomalies in the southern part of the Indian Ocean around the latitude of zero wind stress curl (maximum westerlies), depicted by the contour in Figs. 11 and 12. Importantly, these anomalies extend to the north of the zero wind stress contour, where the climatological wind stress curl is positive, and therefore induce an increase in the equatorward Sverdrup transport. The wind stress curl anomalies are weaker, but still positive when the combined GHG and ozone effect is considered in INTERACT_O$_3$, the ensemble with interactive ozone (Fig. 12c).
They have a similar magnitude to the effect of GHGs in PRESC_O$_3$, the ensemble with prescribed ozone (Fig. 12c), consistent





with both the magnitudes of the westerly winds changes and the magnitudes of the Agulhas leakage changes in these ensembles (Fig. 13). In response to ozone recovery, weak negative wind stress curl anomalies occur that are significant only over limited regions (Figs. 11c and 13e). The Agulhas leakage response is therefore also much weaker compared to the response to increasing GHGs (Fig. 13f).

The relationship between the westerly wind stress and the Agulhas leakage in FOCI is further illustrated in Fig. 14, which shows the decadally-averaged Agulhas leakage relative to 2014-2023 plotted against the decadally-averaged westerly wind stress. Decades of high Agulhas leakage coincide with decades of high westerly wind stress. Furthermore, Fig. 14 also shows that the relatively small changes in wind stress when the GHGs are fixed result in relatively small changes in leakage, while the strongest wind stress intensification that occurs in PRESC_$O_3$, the ensemble with prescribed ozone, results in the largest

increase in leakage. Combined with the good agreement between the changes in the Agulhas leakage and the wind stress curl over the southern part of the Indian Ocean discussed above, this leads us to conclude that ozone recovery and increasing GHGs both impact the Agulhas leakage via their influence on the westerly winds. The two forcings having opposing effects that, when considered together, result in an increase in Agulhas leakage during the twenty-first century.

### 4.2.2   Effects of an increased Agulhas leakage

An enhanced Agulhas leakage has been previously associated with an increased flux of heat and salt into the Atlantic (Biastoch and Böning, 2013). We performed a composite analysis to investigate the relationship between the Agulhas leakage and changes in South Atlantic salinity and temperature in FOCI during the twenty-first century (Fig. 15). In order to exclude any effects of climate change, we based our analysis on the FixGHG ensemble. Anomalies of potential temperature and practical salinity were first obtained for each member of the FixGHG ensemble by removing the 2014-2094 time mean at each location. A 5 year

low-pass filter was applied to the annual Agulhas leakage, temperature and salinity fields to focus on anomalies associated with decadal Agulhas leakage variations and a linear trend was removed from each field. Years of high Agulhas leakage transport were then identified using as threshold the 90th percentile calculated by combining the Agulhas leakage detrended time series of all three members, and insuring that any two years are at least five years apart. This rendered 11 years of high Agulhas leakage over which the detrended anomalies of temperature and salinity averaged over the upper 1000 m were composited

(Fig. 15a, d). Similarly, we composited the temperature and salinity anomalies over a section in the South Atlantic at 35°S between 0° and 20°E (Fig. 15b, c). The significance of the composites was assessed using the Monte Carlo method (e.g., Storch and Zwiers, 1999). A distribution of low-pass filtered and detrended anomalies was obtained by randomly selecting 11 years (the same number as in the original composite) over all three members, without replacement, 1000 times. The anomalies were deemed significant at the 95% level if they were greater than the 97.5th or lower than 2.5th percentile of the distribution.

Figure 15a, d shows that periods of increased Agulhas leakage are associated with simultaneous positive temperature and salinity anomalies spreading from the Cape Basin into the South Atlantic. The depth profiles over the section between 0° and 20°E at 35°S (Fig. 15b, c) reveal that these anomalies are largest in the upper 250 m, but they extend down to 1000 m in the case of temperature and to 750 m in the case of salinity. Five years after a period of high Agulhas leakage, positive salinity anomalies are present in the central part of the South Atlantic, indicating that the salinity anomalies are advected northwestward





towards the western boundary. No significant temperature anomalies can be detected at this lag (not shown), consistent with the idea that thermal anomalies are damped by heat loss to the atmosphere on their way northwards, while salinity anomalies persist, leading to the development of density anomalies (Weijer et al., 2002). Our results support the findings of Biastoch and Böning (2013) and Weijer and van Sebille (2014) that an advective pathway exists for salinity anomalies associated with the Agulhas leakage to the western boundary and into the North Atlantic. This implies that the increase in Agulhas leakage projected to occur during the twenty-first century under the SSP5-8.5 pathway will result in a salinification and densification of the South Atlantic, potentially reaching the North Atlantic, with consequences for the AMOC.

### 4.2.3 Antarctic Circumpolar Current

We now investigate whether the changes in the surface westerly jet and the associated changes in wind stress over the Southern Ocean lead to any changes in the transport of the ACC. Figure 13d shows the changes in the ACC transport calculated from the barotropic streamfunction at two locations, in the Drake Passage and South of Africa. While the Drake Passage is located at the boundary of the high-resolution nest, the region around South Africa is located in the middle of the nest. Following Durgadoo et al. (2013), the ACC transport through the Drake Passage is computed as the difference in the barotropic streamfunction between the Antarctic Peninsula ($60°$W and $65°$S) and South America ($68°$W and $55°$S), while the ACC transport South of Africa is computed as the maximum in the barotropic streamfunction between $20°$E and $30°$E. The mean current-day (2011-2030) transport through the Drake Passage is 97.9 Sv and 95.3 Sv in FOCI with interactive ozone chemistry (INTERACT_$O_3$) and with prescribed CMIP6 ozone (PRESC_$O_3$), respectively. This is at the lower end of the CMIP5 range of $90 - 264$ Sv (Meijers et al., 2012) and lower than the observed transport, which ranges from $136.7 \pm 7.8$ Sv (Cunningham et al., 2003) to $173.3 \pm 10.7$ Sv (Donohue et al., 2016), although these measurements cover earlier time periods. South of Africa, the current-day ACC transport is 181.8 Sv and 181.2 Sv in INTERACT_$O_3$ and PRESC_$O_3$, respectively.

Ozone depletions results in a weak but statistically significant (at the 95% level) reduction of the ACC transport at both locations, in line with the weakening of the westerlies. This weakening represents 2% and 1% of the current-day mean ACC transport in the Drake Passage and South of Africa, respectively. In contrast, increasing GHGs lead to a strengthening of 6% and 4% of the current-day mean ACC transport in the Drake Passage and South of Africa, respectively, as expected from the strengthening of the westerlies. There is a larger spread in the ACC transport changes in the individual simulations South of Africa compared to the Drake Passage, as the former region is located in the middle of the high-resolution nest and is therefore characterized by more vigorous eddy activity. Nevertheless, the magnitude of the ensemble mean changes are in good agreement at the two locations for all of the ensembles, giving confidence in the results. Interestingly, the combined effect of ozone recovery and increasing GHGs on the ACC is very similar in magnitude to the effect of increasing GHGs alone. The ACC transport in the Drake Passage and South of Africa is stronger by 6.0 Sv and 6.9 Sv, or 6% and 4% , respectively at the end of the twenty-first century compared to today. This points to a non-linear response to the combined forcing. It is known that changes in the ACC transport do not scale with the increase in the wind stress over the Southern Ocean due to eddy saturation (e.g., Farneti et al., 2010). The stronger wind stress increases the northward Ekman transport over the Southern Ocean resulting in a steepening of isopycnals across the ACC, on one hand, and it increases eddy activity, which acts to flatten





isopycnals, on the other hand. The two effects offset each other, resulting in only small changes in the transport of the ACC.

The ACC is closer to perfect saturation for stronger wind stress forcing and it seems that a threshold is reached after which a further increase in the wind stress does not lead to a further increase in the ACC transport. Eddy saturation can therefore explain the similar increase in the ACC transport in the absence and in the presence of ozone recovery, despite a stronger increase in the wind stress associated with the westerlies, when ozone recovery is excluded. Another possible explanation is the large spread in the density gradient across the ACC seen in the FOCI simulations, which results in a large spread of ACC

transport estimates.

The ACC transport experiences a significantly stronger increase during the twenty-first century when ozone is prescribed (PRESC_$O_3$) compared to when ozone is calculated interactively (INTERACT_$O_3$; Fig. 13d). This is consistent with the fact that the westerlies strengthen more in PRESC_$O_3$ than in INTERACT_$O_3$ (Figs. 12b, e and 13c), but may seem at odds with the eddy saturation phenomenon. However, the 2011-2030 mean westerlies strength in the band $45°S - 60°$ is lower

in PRESC_$O_3$ than in INTERACT_$O_3$, 5.74 m s$^{-1}$ compared to 5.96 m s$^{-1}$, respectively. The ACC is therefore less eddy saturated in PRESC_$O_3$ and stronger changes in transport can occur. Consistent with the weaker winds, the current-day mean ACC transport is also weaker in PRESC_$O_3$ than in INTERACT_$O_3$, as noted above. This shows that the choice of the ozone field affects both the mean ACC transport and its response to ozone recovery and increasing GHGs and should be carefully considered in studies investigating past and future ACC transport changes. To further highlight the importance of the ozone

field, we also note that the difference between the increase in the ACC transport in PRESC_$O_3$ and INTERACT_$O_3$ is slightly larger in magnitude than the whole effect of ozone recovery (Fig. 13d).

## 5 Discussion

### 5.1 Ozone recovery

The October Antarctic TCO in austral spring returns to 1980s values in 2048 in FOCI under the SSP5-8.5 pathway. While

this is sooner than estimates from the CCMI ensemble (Dhomse et al., 2018; Amos et al., 2020), the earlier return date can be explained by the higher levels of $CO_2$ and $CH_4$ in the emission scenario used in this study, as these GHGs accelerate ozone recovery over Antarctica. In the upper stratosphere, the increase in GHGs contributes about half of the total ozone recovery over the twenty-first century in FOCI. This is explained by the GHG-induced cooling, which slows down the reactions involved in destroying ozone, as shown by Haigh and Pyle (1982); Rosenfield et al. (2002); Jonsson et al. (2004); Revell et al. (2012).

At lower stratospheric levels, GHGs cause an increase in ozone through transport changes associated with an enhanced BDC.

### 5.2 Temperature and zonal wind

FOCI simulates a warming of the spring lower stratosphere over Antarctica accompanied by a weakening of the polar vortex due to ozone recovery, and an overall cooling of the stratosphere, warming of the troposphere and eastward acceleration of the tropospheric and stratospheric winds throughout the year due increasing GHGs. These results are in good agreement with



the findings of previous studies (Perlwitz et al., 2008; Karpechko et al., 2010; McLandress et al., 2010; Polvani et al., 2011a).
We extended these findings by analyzing for the first time the spatial patterns of changes during austral spring. Our analysis
revealed that GHGs have a strong impact on the spring lower stratospheric temperature over Antarctica, inducing changes
almost as large as those resulting from the recovery of the ozone hole. By altering the vertical propagation of wave activity and
therefore the residual circulation in the stratosphere, GHGs warm the lower stratosphere over West Antarctica and the Pacific

sector of the Southern Ocean, but cool the lower stratosphere over East Antarctica and the Indo-Atlantic sectors. The increase in
GHGs also leads to a shift of the spring polar vortex towards the Indian Ocean, strengthening the outer flank of the vortex in this
region and weakening the outer flank over the Pacific sector. This dipole structure characteristic of the GHG-induced austral
spring temperature and zonal wind changes results in an underestimation of the spring GHG effect when zonal or areal mean
fields are analyzed. FOCI simulates a statistically insignificant warming in the lower stratosphere during October when the

temperature change due to GHGs is averaged over the polar cap, while Karpechko et al. (2010) reported a significant warming
of the Antarctic lower stratosphere in their ensemble forced solely by increasing GHGs. In contrast, Son et al. (2008, 2009)
found a weak cooling of the Antarctic lower stratosphere in the CMIP3 models that exclude ozone recovery in agreement with
McLandress et al. (2010), who also found that GHGs drive a weak polar cap lower stratospheric cooling in their CCM. These
contrasting results can be explained by intermodel differences in simulating the strength of the GHG-induced change within

each pole, which lead to different levels of compensation between the warming and the cooling pole. In contrast, the effect of
ozone recovery is a warming of the entire polar cap and a circumpolar weakening of the stratospheric westerlies. Therefore,
there is good agreement among studies regarding the direction of ozone-induced twenty-first century stratospheric changes
(Perlwitz et al., 2008; Karpechko et al., 2010; McLandress et al., 2010).

## 5.3    The Brewer-Dobson circulation and wave activity

We further examined the impacts of ozone recovery and increasing GHGs on the BDC. Although the reduction of ODSs could
also theoretically lead to changes in the BDC through the radiative effects of the ODSs, Abalos et al. (2019) showed that trends
in the BDC due to ODSs are mainly explained by ozone changes. We therefore attribute the ODS-induced BDC changes in our
simulations to ozone recovery. Our results add to the body of evidence (Oman et al., 2009; McLandress et al., 2010; Lin and
Fu, 2013; Oberländer et al., 2013; Polvani et al., 2018, 2019) that the consequence of ozone recovery during the twenty-first

century will be to weaken the BDC and therefore partly offset the strengthening induced by the increase in GHGs. All of the
above-mentioned studies reported contrasting effects on the BDC in austral summer due to the two forcings. While we also
find that GHGs strengthen and ozone recovery weakens the BDC in the SH during summer, the summer changes are small
when compared to springtime changes and, for the case of ozone recovery, they are dominated by changes during December.
This result is in line with the presence of climatological easterlies in the middle and upper stratosphere throughout most of the

summer that do not support Rossby wave propagation and therefore confine the changes in wave drag to the lower stratosphere,
where the zero wind line is located. Our monthly analysis revealed that ozone recovery has the largest impact on the BDC in
the SH between October and December. It leads to a reduction of the polar cap downwelling that starts in the upper stratosphere
in October and propagates downward in the following months and to a weak but significant strengthening of the downwelling





in the lower stratosphere in September and October. The latter result confirms the finding of McLandress et al. (2010) that the trends in the polar downwelling due to ozone recovery change sign between spring and summer in the lower stratosphere. In our simulations, however, the change in sign occurs already in November. This offers a possible explanation as to why previous studies that looked at seasonally averaged changes did not find an impact of ozone recovery on the BDC in spring (Polvani et al., 2018; Lin and Fu, 2013). The ozone-induced changes in the polar cap downwelling are caused by shifts in the height at which wave drag is being deposited as a result of wave breaking, related to the weakening and the earlier breakdown of the polar vortex caused by ozone recovery. This mechanism is analogous to that demonstrated to be responsible for mediating the effect of increasing GHGs on the BDC in the subtropical lower stratosphere by Shepherd and McLandress (2011).

Increasing GHGs lead to a weakening of the shallow BDC branch in November, amplifying the effect of ozone recovery and contrasting the strengthening effect of GHGs during the rest of the year. In fact, our three-dimensional analysis revealed that, during austral spring, GHGs have opposite impacts on the flux of wave activity and therefore on the BDC on different sides of the Antarctic continent, which represents a novel result. Closely related to the GHG-induced changes in stratospheric winds, whereby the polar vortex shifts towards the Indian Ocean, the flux of wave activity is enhanced over the Indian sector and weakened over the Pacific sector in October. Consequently, the downwelling above the Indian Ocean is enhanced in the upper stratosphere and weakened in the lower stratosphere, while the opposite occurs above the Pacific sector, driving dynamical heating changes of similar magnitude to the radiative effect of ozone recovery. The changes in the polar vertical residual velocity reveal that the Indian sector dominates in the zonal mean during October above 20 hPa and the polar cap downwelling strengthens. In November, the Pacific sectors becomes dominant, wave breaking occurs even lower in the stratosphere as the polar vortex breaks down, and the polar downwelling weakens in the lower stratosphere, together with the entire lower branch of the SH BDC. An important result revealed by our three-dimensional analysis is that the GHG-induced changes in the residual circulation as inferred from the TEM framework are dictated by the degree of compensation between the two regions of contrasting change. Therefore, the zonal mean fields do not show the complete picture of the austral spring GHG influence on the BDC, which exhibits strong zonal variations. The compensation between the two regions likely varies among different models. This would explain why previous studies that investigated austral spring BDC changes due to increasing GHGs disagree on the direction of change. While Karpechko et al. (2010) reported a strengthening of the polar downwelling in the lower stratosphere, McLandress and Shepherd (2009); McLandress et al. (2010) reported a weakening and Polvani et al. (2018) found no significant change. This hypothesis needs to be tested in future studies involving a large number of climate models. The reasons behind the different regional impacts of GHGs in austral spring and behind the timing of these opposing impacts are currently not known and also deserves further investigation.

## 5.4 Surface westerly winds and SAM

Our analysis of the twenty first changes in the SH surface westerly winds supports previous findings (Shindell and Schmidt, 2004; Yin, 2005; Perlwitz et al., 2008; McLandress et al., 2011; Polvani et al., 2011a) that ozone recovery drives a weakening and equatorward shift of the westerlies during austral summer accompanied by a negative change in the SAM, while increasing GHGs drive a year-round strengthening and poleward shift accompanied by a positive change in the SAM. Under the high





emission scenario SSP5-8.5, FOCI projects that the GHG effect overwhelms the effect due to ozone recovery both during austral summer and in the annual mean, such that the westerlies shift farther poleward and the SAM shifts more towards its

positive phase. This result is consistent with previous projections for the same high emission scenario (Bracegirdle et al., 2020) or for the similar scenario RCP8.5 (Eyring et al., 2013; Barnes et al., 2014; Gerber and Son, 2014; Iglesias-Suarez et al., 2016; Tim et al., 2019). However, it contrasts projections for low or moderate emission scenarios. CCMs and atmospheric models with prescribed ozone forced by the moderate CMIP3 A1B scenario predict either a dominance of the ozone effect and a summertime decrease in the SAM and equatorward shift of the westerlies (Perlwitz et al., 2008; Son et al., 2008) or

a compensation of the ozone and GHG effects and consequently no significant changes in the SAM or the westerlies during austral summer (Son et al., 2010; McLandress et al., 2011; Polvani et al., 2011a; Gerber and Son, 2014). The latter result is supported by simulations with coupled climate models forced with the same scenario (Son et al., 2008, 2010; Gerber and Son, 2014), the CMIP5 moderate scenario RCP4.5 (Eyring et al., 2013; Barnes et al., 2014; Gerber and Son, 2014), or the CMIP6 moderate scenario SSP2-4.5 (Bracegirdle et al., 2020). Under the low emission scenarios RCP2.6 for CMIP5 and SSP1-2.6 for

CMIP6, coupled climate models and CCMs predict a negative SAM trend and an equatorward shift of the westerlies (Barnes et al., 2014; Eyring et al., 2013; Iglesias-Suarez et al., 2016; Tim et al., 2019; Bracegirdle et al., 2020). This highlights that the changes in the SH westerlies during the rest of the century will strongly depend on the future emissions of GHGs.

## 5.5 Agulhas leakage

The projected poleward intensification of the surface westerlies under SSP5-8.5 leads to pronounced ocean circulation changes.

The flux of warm and saline waters from the Indian into the Atlantic Ocean, known as the Agulhas leakage, is increasing during the twenty-first century under combined ozone and GHG forcing, by 13% in the ensemble with interactive ozone and by 33% in the ensemble with prescribed ozone relative to the 2014-2028 mean. The increase in Agulhas leakage is associated with increased temperature and salinity in the southeastern South Atlantic. Our results are consistent with those of Biastoch et al. (2015) and Cheng et al. (2018), who related an increased Agulhas leakage to higher SSTs extending from the Cape Basin

into the South Atlantic. We additionally showed that this relationship is also valid at depth, down to 1000 m. In contrast with the two afore-mentioned studies, we do not detect negative temperature anomalies to the east of the Agulhas Retroflection during periods of high Agulhas leakage. This relationship occurs only on interannual time scales and disappears if longer time scales are considered, as in the present study. The link between increased Agulhas leakage and higher southeastern Atlantic temperatures simulated in FOCI in the upper 1000 m supports the findings of Lübbecke et al. (2015) that the positive Agulhas

leakage trend in the second half of the twentieth century contributed to the warming of the eastern tropical Atlantic. However, the temperature anomalies are not detectable anymore in FOCI after 5 years. This could be partly explained by loss of heat to the atmosphere that damps the temperature anomalies, as reported by Weijer et al. (2002) and Biastoch and Böning (2013). The salinity anomalies, in contrast, are advected northwestward and can be detected in the central part of the South Atlantic at a lag of 5 years. This results in the development of density anomalies that are advected towards the North Atlantic and

that can potentially be important for the stability of the AMOC, as they have the ability to partially compensate the changes in density resulting from a freshening of the North Atlantic due to climate change-induced ice sheet melting. Unfortunately, in the





current study we cannot isolate the impact of the increase in Agulhas leakage on the AMOC from other effects of the increasing GHGs. Nevertheless, there is compelling evidence from previous studies that salinity anomalies related to an increased Agulhas leakage reach the subtropical North Atlantic within two to four decades (Biastoch and Böning, 2013; Weijer and van Sebille, 2014). In addition, 40% of the Agulhas leakage was found to reach 24°N within two decades (Rühs et al., 2013). Therefore, the twenty-first century increase in Agulhas leakage is likely to alter the thermohaline structure of the AMOC.

### 5.6 Antarctic Circumpolar Current

A further consequence of the changes in the SH westerly winds is a change in the transport of the ACC during the twenty-first century. We showed that ozone recovery drives a statistically significant decrease of 2% (1%) in the ACC transport through the Drake Passage (south of Africa) and that increasing GHGs drive an increase in the ACC transport of 6% (4%) at the end of the twenty-first century compared to the current day, in agreement with the findings of Sigmond et al. (2011). However, the combined effect of ozone recovery and increasing GHGs equals the effect of GHGs alone, with the ACC transport increasing by 6% (4%) in the Drake Passage (south of Africa). This likely occurs due to the well-known eddy saturation phenomenon (e.g., Farneti et al., 2010; Morrison and Hogg, 2013), whereby the effects of an increase in eddy activity and of an increase in northward Ekman transport offset each other rendering the ACC (almost) insensitive to changes in wind stress. Typically, coupled climate models such as those used for the different phases of CMIP have too low a resolution to simulate mesoscale eddies and rely on eddy parametrization schemes based on Gent and Mcwilliams (1990) to account for eddy effects (e.g., Downes and Hogg, 2013). CMIP5 models do not agree on the sign of the change in ACC transport during the twenty-first century under RCP8.5, with about half of the models simulating an increase and the other half simulating a decrease in transport (Downes and Hogg, 2013). Additionally, there is no statistically significant relationship between the change in the strength of the zonal wind stress and the change in ACC transport among the CMIP5 models (Meijers et al., 2012; Downes and Hogg, 2013). The ACC transport changes in the Drake Passage of about 6% and 11% in FOCI with interactive ozone and with prescribed ozone, respectively, fall within the range of changes of about ± 15% simulated by CMIP5 models under RCP8.5 (Downes and Hogg, 2013). Unlike the CMIP5 models, however, FOCI has a higher oceanic resolution, 1/2° globally and 1/10° within the high resolution nest, INALT10X. The mesoscale eddies are resolved in the Atlantic sector and in part of the Indian sector of the Southern Ocean and because of the two-way nesting technique used in FOCI, the information about the presence of the eddies is transmitted to the global ocean model. Therefore, we can assume that this configuration of FOCI captures the eddy saturation effect better than the CMIP-class models. Bishop et al. (2016) performed a perturbation experiment with another coupled climate model that uses a 1/10° ocean in which they increased the zonal wind stress over the Southern Ocean by 50% and found only a 6% strengthening in the ACC transport through the Drake Passage in response. In our simulations, a 6% and an 11% ACC transport increase is obtained for the wind stress increases of 10% and 17% in INTERACT_$O_3$ and PRESC_$O_3$, respectively, implying that the sensitivity of the ACC transport to wind stress changes is stronger in our model. Although our results are not directly comparable with those of Bishop et al. (2016) because their simulation does not include changes in buoyancy due to, for example, climate change, this suggests that the degree of eddy saturation is further increased if the oceanic resolution is eddy-resolving over the entire Southern Ocean.





### 5.7 Importance of the ozone field

The comparison of the combined ozone and GHG effect between the ensemble with interactive ozone chemistry, INTERACT_$O_3$, and the ensemble in which the CMIP6 ozone field is prescribed, PRESC_$O_3$, revealed that the ozone effect is weaker and the GHG effect is more dominant when the ozone field is prescribed. This hints to the need to include interactive ozone chemistry

to realistically represent all ozone-related processes relevant for predictions of twenty-first century climate. Ideally, the result that the ozone effect is weaker when the ozone field is prescribed should be verified in future studies by comparing the CMIP6 models with interactive chemistry with the same models, but with prescribed CMIP6 ozone. In FOCI, significant differences were found in all the examined fields. In the stratosphere, differences in both the magnitude and the spatial pattern of the total response occur. The magnitude of the difference at the surface and in the oceanic circulation is as large as the ozone effect

itself. The combined effect when the ozone field is prescribed is in some cases larger than the GHG effect simulated alone in the ensemble in which the ozone field is calculated interactively. This suggests that the GHG effect is affected by the response of the ozone field and is greater when the ozone is prescribed. The studies by Morgenstern et al. (2014) and Chiodo and Polvani (2016) showed that there are important dynamical changes resulting from the response of the ozone field to GHGs and that these changes offset part of the direct influence of GHGs on the SH dynamics. Although the prescribed CMIP6 ozone field is

consistent with the increase in GHGs under SSP5-8.5, it cannot react to the changes in GHG concentrations during the simulations to which it is prescribed and may therefore be less sensitive to the GHG changes than in the ensemble in which it evolves interactively. This, in turn, could result in a lower degree of compensation between the direct GHG effect on the dynamics and the GHG effect mediated by the ozone field. In a similar manner, the ozone field in PRESC_$O_3$ does not react to the simulated changes in temperature and dynamics. It is therefore important to note that the combined effect in the ensemble with prescribed

ozone should not be viewed as the sum of the individual ozone and GHG effects from the ensembles with interactive ozone.

    The increase in the Antarctic polar cap ozone during the twenty-first century is larger in FOCI with interactive ozone than in the CMIP6 ozone field and, in addition, the spatial pattern of the increase differs. This can explain a part of the differences between INTERACT_$O_3$ and PRESC_$O_3$, but given the fact that the differences in some fields are as large as the impact of ozone recovery, it is likely not the only cause. Another source of differences could be the temporal interpolation of the monthly

CMIP6 ozone field, which leads to an underestimation of the fast springtime ozone changes, as shown by (Sassi et al., 2005; Neely et al., 2014). In addition, the lack of consistency between the prescribed ozone field and the simulated dynamics can also lead to weaker ozone effects, as argued by Ivanciu et al. (2021).

    The differences in the circulation response to ozone recovery and increasing GHGs between INTERACT_$O_3$ and PRESC_$O_3$ show that the choice of the ozone field can affect the prediction of changes not only in the atmospheric, but also in the oceanic

circulation. This adds to the evidence provided by (Li et al., 2016) and Seviour et al. (2016) that the ozone field affects the SH oceanic circulation. Furthermore, it suggest that the ozone field may introduce a source of uncertainty in the predictions for the twenty-first century.





## 6 Summary

The unique ocean-atmosphere coupled climate model FOCI with interactive ozone chemistry and enhanced ocean resolution
was used in this study to separate the effects of ozone recovery and increasing GHGs on the atmospheric and oceanic circulation
in the SH during the twenty-first century, under the high emission scenario SSP5-8.5. A special emphasis was placed on the
spatial patterns that characterize these effects. In addition, the combined effects of the two forcings were compared between
the configuration of FOCI with interactive ozone chemistry and the configuration in which the CMIP6 ozone field consistent
with the SSP5-8.5 pathway was prescribed. Our main findings are:

 – The springtime Antarctic total column ozone in FOCI returns to 1980s levels in 2048, as increasing GHGs following
   SSP5-8.5 accelerate the recovery of the ozone hole by about two decades.

 – Ozone recovery during the twenty-first century reverses the effects of the past ozone depletion in agreement with previous
   studies (e.g., McLandress et al., 2010), leading to a warming of the lower stratospheric polar cap in austral spring and a
   weakening of the westerly winds both in the stratosphere and in the troposphere. In contrast, increasing GHGs drive a
cooling of the stratosphere throughout most of the year and westerly wind anomalies peaking at the top of the subtropical
   jet and extending vertically to the surface and to the top of the stratosphere.

 – GHGs have an important dynamical impact on the SH stratosphere during austral spring that differs from their effect
   during the rest of the year. This zonally asymmetric effect has been previously overlooked. GHGs drive temperature and
   zonal wind anomalies of opposite sign over the Pacific and Indo-Atlantic sectors of the Southern Ocean, which are of
similar magnitude to the ozone-induced anomalies. Therefore, the ozone and GHG effects reinforce each other over the
   Pacific sector and counteract each other over the Indo-Atlantic sector.

 – Associated with the GHG-induced temperature and zonal wind changes, the propagation of wave activity during austral
   spring is altered by the increasing GHGs. Over the Eastern Hemisphere in October, planetary waves can propagate
   higher into the stratosphere and wave breaking is suppressed (enhanced) in the middle (upper) stratosphere, leading
to a weakening (strengthening) of the polar downwelling and dynamical cooling (warming). The opposite occurs over
   the Western Hemisphere. These results highlight that the increasing GHGs play an important role in the dynamical
   stratospheric changes that take place in austral spring during the twenty-first century. The changes in the polar zonal
   mean residual vertical velocity derived within the TEM framework are dictated by the changes in the Eastern Hemisphere
   in October and by the changes in the Western Hemisphere in November. As the dynamical response to increasing GHGs
exhibits strong zonal variations, the changes in zonal mean fields, which have been investigated in previous studies, do
   not fully capture this GHG effect.

 – The flux of wave activity is reduced during spring by ozone recovery, leading to a weakening of the polar downwelling in
   the upper stratosphere in October and throughout the stratosphere in November and December. The resulting dynamical
   cooling diminishes the warming that occurs due to the radiative effect of ozone in the lower stratosphere and causes an
overall cooling of the upper stratosphere in these months.





- The increase in GHGs drives a strengthening of the SH BDC throughout most of the year, with the exception of November, when it leads to a weakening instead, reinforcing the weakening effect of ozone recovery during this month. Therefore, the combined effect of ozone recovery and increasing GHGs on the BDC is a weakening in November and December and a strengthening during the rest of the year.

- At the surface, ozone recovery drives a shift towards the negative phase of the SAM and a poleward weakening of the westerly winds that occur in late spring and summer and that are significant in the annual mean. The increase in GHGs has an opposing effect, driving a poleward intensification and shift of the westerlies, accompanied by a positive change in the SAM. The GHG effect dominates, but the surface changes are weaker when both forcings are considered than those driven by GHGs alone.

- The Agulhas leakage, which transports warm and saline waters from the Indian Ocean into the Atlantic Ocean, is affected by both increasing GHGs and ozone recovery through their impact on the SH westerlies. Ozone recovery leads to a weak reduction in leakage, while the increasing GHGs lead to an increase in leakage that dominates when both forcings are considered together, consistent with the changes in the westerly winds. The resulting increase in Agulhas leakage is associated with positive temperature and salinity anomalies in the southeastern part of the South Atlantic that extend to about 1000 m depth. While the temperature anomalies are damped by heat loss to the atmosphere, the salinity anomalies are advected northwestwards and have the potential to reach the North Atlantic and to influence the stability of the AMOC.

- The transport of the ACC in the Drake Passage and south of Africa is also impacted by ozone recovery and by climate change, but changes in transport are rather small, with a maximum of 6%. Although ozone recovery drives a small but significant weakening of the ACC transport at both locations and GHGs drive an increase in transport, when both forcings are considered together the transport increases by an amount similar to that due to GHGs alone. This result is consistent with the fact that the ACC is in an eddy saturated state and larger wind changes do not result in larger changes in the ACC transport.

- There are significant differences in the total effect of ozone recovery and increasing GHGs between the ensemble with interactive ozone and the ensemble with prescribed ozone. In the stratosphere, when the CMIP6 ozone field is prescribed, the total response of the polar cap temperature, of the westerly winds and of the BDC is weaker during the months and at the locations where ozone recovery plays an important role (e.g. spring in the lower and middle stratosphere). In addition, the two ensembles exhibit spatial differences in the response of the SH dynamics to the combined forcing. At the surface and in the ocean, where the GHG effect is driving the changes, the ensemble with prescribed ozone simulates a stronger total response. All these differences show that the effects of ozone recovery are significantly weaker when the CMIP6 ozone field is prescribed. This is likely due to a combination of the following factors: the increase in the Antarctic polar cap ozone is stronger in our model with interactive ozone chemistry than in the CMIP6 ozone field, the prescribed ozone field is not consistent with the evolution of the dynamics in the model, the feedbacks between the ozone field, radiation



and dynamics cannot occur if the ozone field is prescribed and are therefore not properly captured, and the interpolation
of the prescribed monthly-mean ozone field in time, which causes an underestimation of the spring ozone changes. The
magnitude of the difference between the ensemble with interactive ozone and the ensemble with prescribed ozone in
the changes experienced by the SAM, the surface westerlies, the Agulhas leakage and the ACC transport is comparable
to or greater than the magnitude of the ozone effect itself. This highlights the importance of the choice of the ozone
field in studies that investigate the future changes in both the atmospheric and the oceanic circulation. Furthermore, it
suggests that the choice of the ozone field is a source of uncertainty in the predictions for the twenty-first century. While
including interactive ozone chemistry is the only way to ensure that all ozone-related processes are simulated and that
the ozone field is spatially and temporally consistent with the simulated dynamics, there is no way of knowing which
of the simulated changes are closer to reality. Even for the historical period for which observations are available, the
comparison of model results with the real world is further complicated by model biases and by inter-model differences.
For example, if a model is characterized by a too strong polar cap cooling, the inclusion of interactive ozone would result
in an even larger cooling and therefore exacerbate this bias, although it offers the more realistic representation of ozone
processes.

While in the stratosphere the effects of ozone recovery and increasing GHGs reinforce each other in some regions and coun-
teract each other in other regions, the GHG effect clearly dominates the changes at the surface and in the oceanic circulation
under the high emission scenario used here. However, these changes are considerably weaker in the presence of ozone recovery
than in its absence, highlighting the importance of the Montreal Protocol in mitigating some of the impacts of climate change.

*Code and data availability.* The output of the model simulations used in this study can be found at https://doi.org/10.5281/zenodo.5013716
(Ivanciu, 2021). The scientific code used to perform the analysis can be obtained upon request from Ioana Ivanciu (iivanciu@geomar.de).

*Author contributions.* II, KM and AB designed the study and the set up of the simulations. II, SW and JH performed the model simulations.
II carried out the analysis and all authors discussed the results. II wrote the manuscript with contributions from all co-authors.

*Competing interests.* The authors declare that they do not have any conflict of interest.

*Acknowledgements.* This study has been funded by the German Federal Ministry of Education and Research through the SPACES-II CA-
SISAC project (grant no. 03F0796A). The model simulations used in this study were performed with resources provided by the North-
German Supercomputing Alliance (HLRN). The authors thank Siren Rühs and Franziska Schwarzkopf for helpful discussions regarding the
analysis of the surface and oceanic circulation.



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



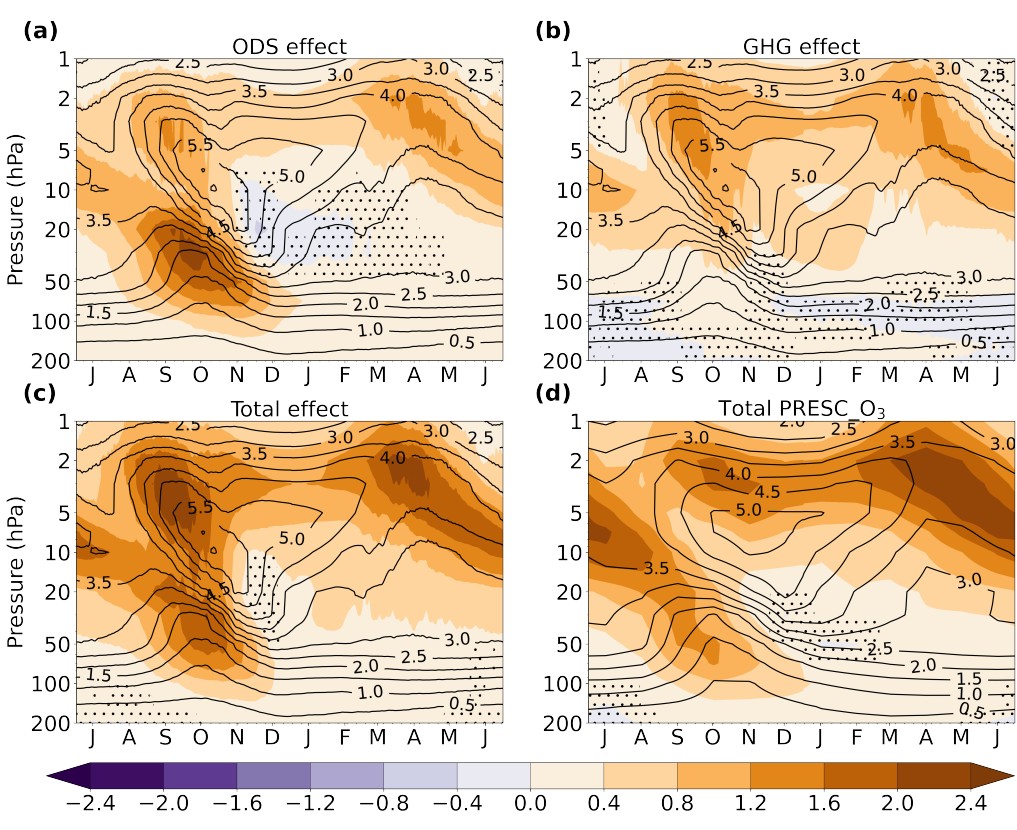

**Figure 1.** Changes in the polar cap (70°S-90°S) ozone (ppmv) for each day of the year and pressure level (color shading): effect of ODSs (a), effect of GHGs (b), total effect when ozone is interactive (c) and total effect prescribed in PRESC_O$_3$ (d). The contours depict the current day (2011-2030) climatology from INTERACT_O$_3$ in a-c and prescribed in PRESC_O$_3$ in d. The stippling masks regions where the changes are not significant at the 95% confidence interval based on a two-tailed t-test.

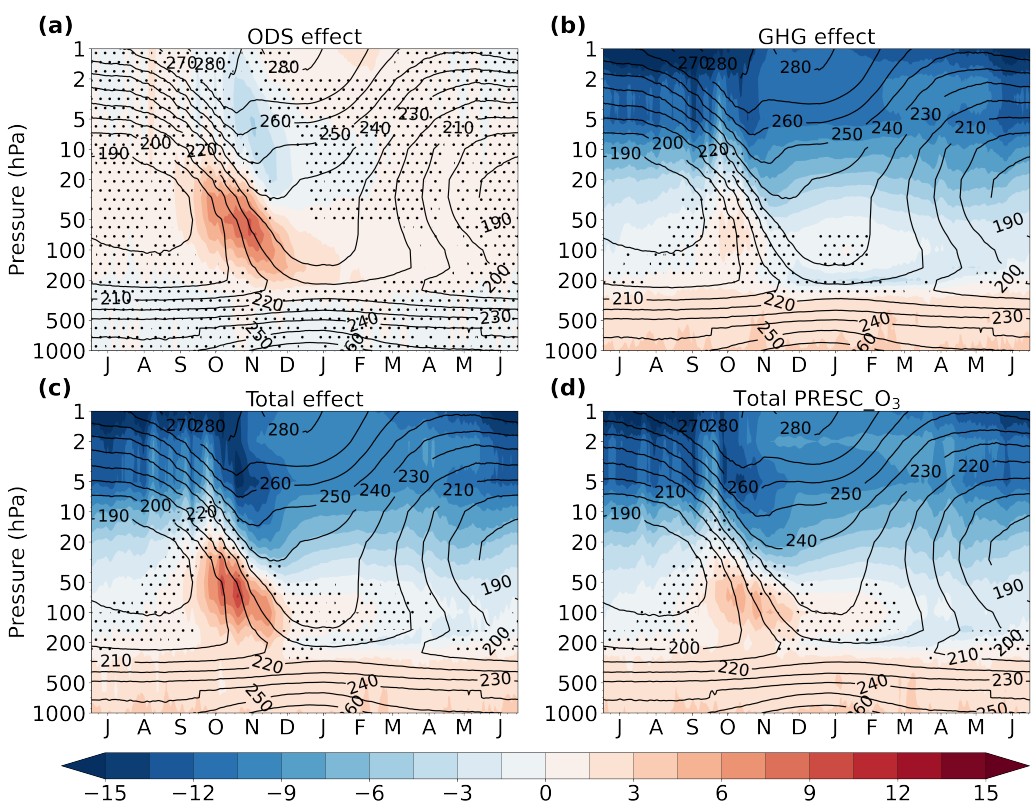

**Figure 2.** Changes in the polar cap (70°S-90°S) temperature (K) for each day of the year and pressure level (color shading): effect of ozone recovery (a), effect of GHGs (b), total effect in INTERACT_$O_3$ (c) and total effect in PRESC_$O_3$ (d). The contours depict the current day (2011-2030) climatology from INTERACT_$O_3$ in a-c and from PRESC_$O_3$ in d. The stippling masks regions where the changes are not significant at the 95% confidence interval based on a two-tailed t-test.



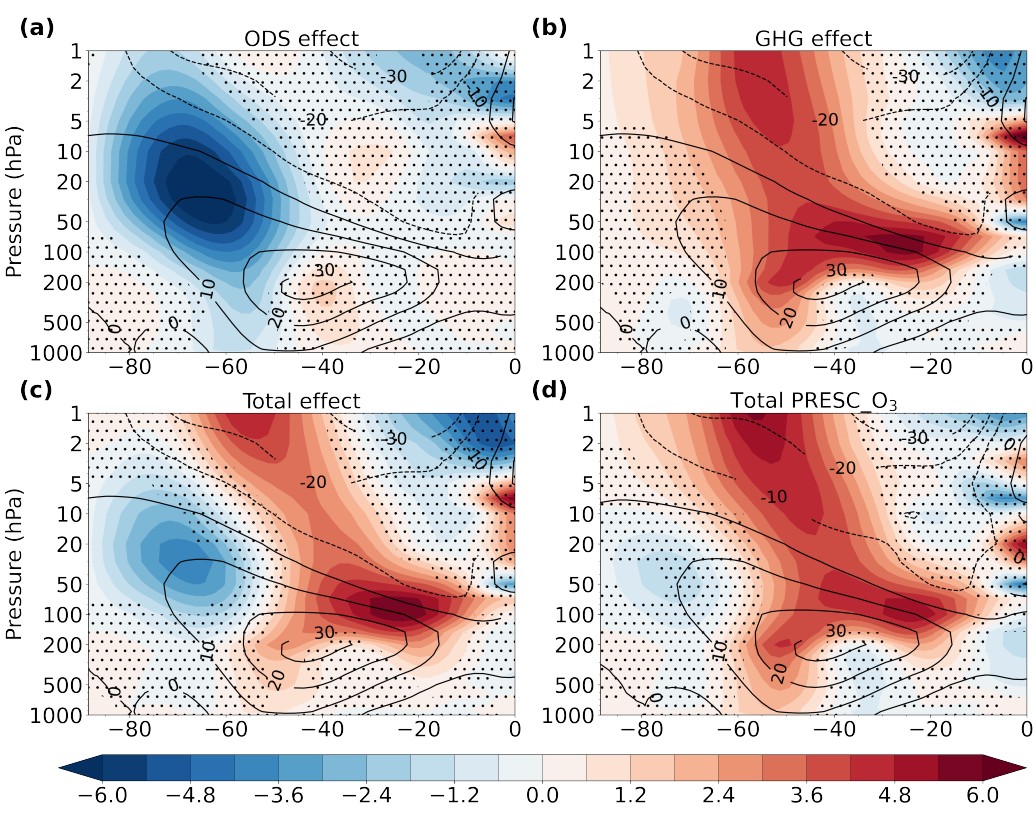

**Figure 3.** Changes in the November-December zonal mean zonal wind (m s$^{-1}$) for each latitude and pressure level (color shading): effect of ozone recovery (a), effect of GHGs (b), total effect in INTERACT_O$_3$ (c) and total effect in PRESC_O$_3$ (d). The contours depict the current day (2011-2030) climatology from INTERACT_O$_3$ in a-c and from PRESC_O$_3$ in d. The stippling masks regions where the changes are not significant at the 95% confidence interval based on a two-tailed t-test.



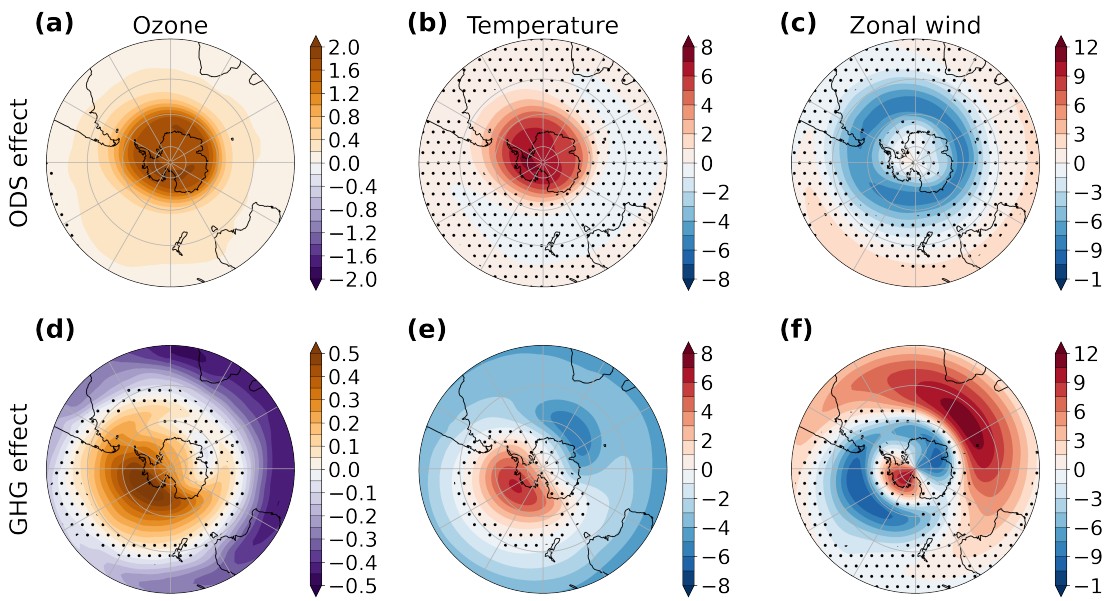

**Figure 4.** Changes in ozone (a and d; ppmv) and temperature (b and e; K) at 50 hPa and in zonal wind (c and f; m s$^{-1}$) at 20 hPa during October due to ODSs (a-c) and GHGs (d-f). The stippling masks regions where the changes are not significant at the 95% confidence interval based on a two-tailed t-test.



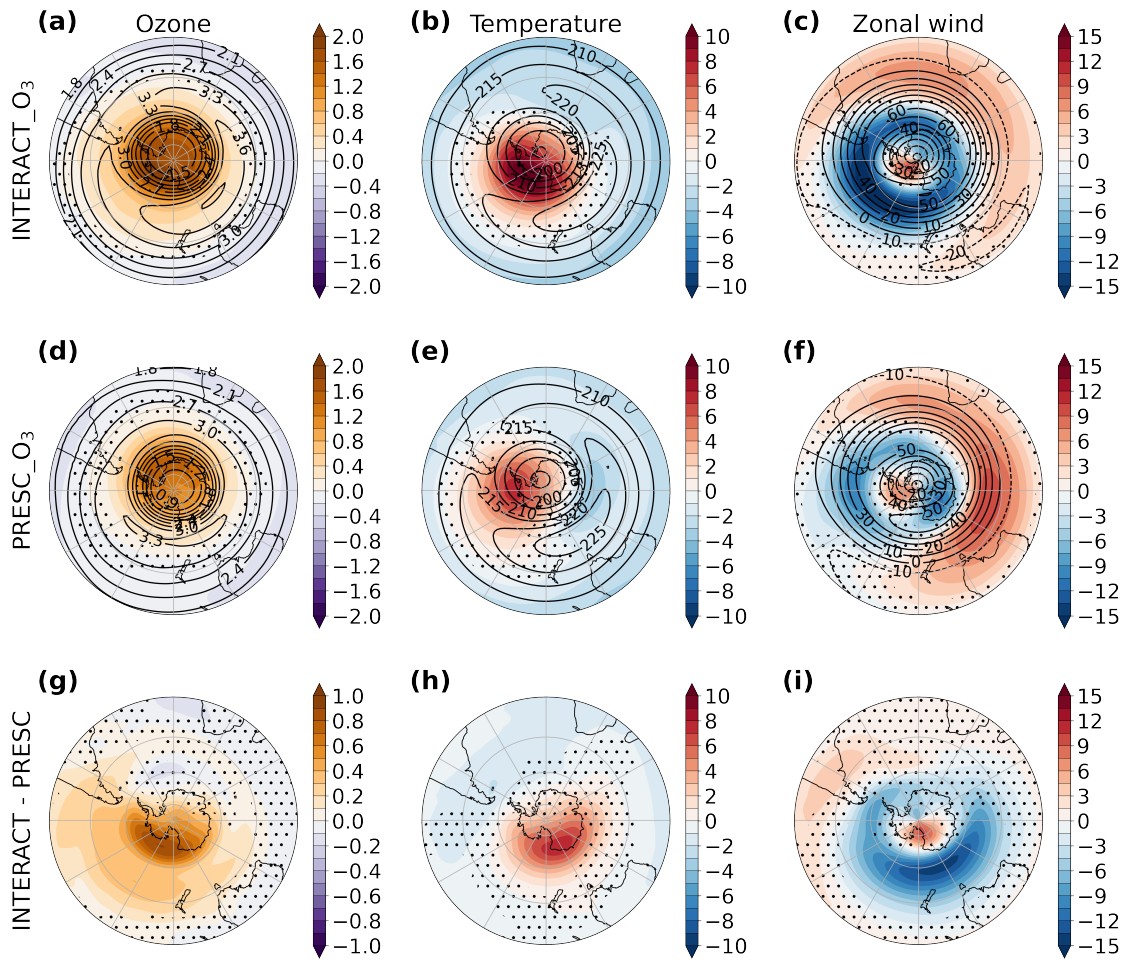

**Figure 5.** Changes in ozone (a and d; ppmv) and temperature (b and e; K) at 50 hPa and in zonal wind (c and f; m s$^{-1}$) at 20 hPa due to ODSs and GHGs combined, in INTERACT_O$_3$ (a-c) and PRESC_O$_3$ (d-f), and the difference between the INTERACT_O$_3$ and the PRESC_O$_3$ changes (g-i). The stippling in a-f masks regions where the changes are not significant at the 95% confidence interval based on a two-tailed t-test, and the stippling in g-i masks regions where the difference is lower than $2\sigma$ inferred from the spread of the individual ensemble members. The contours in a-f show the current-day (2011-2030) climatologies in the respective fields.

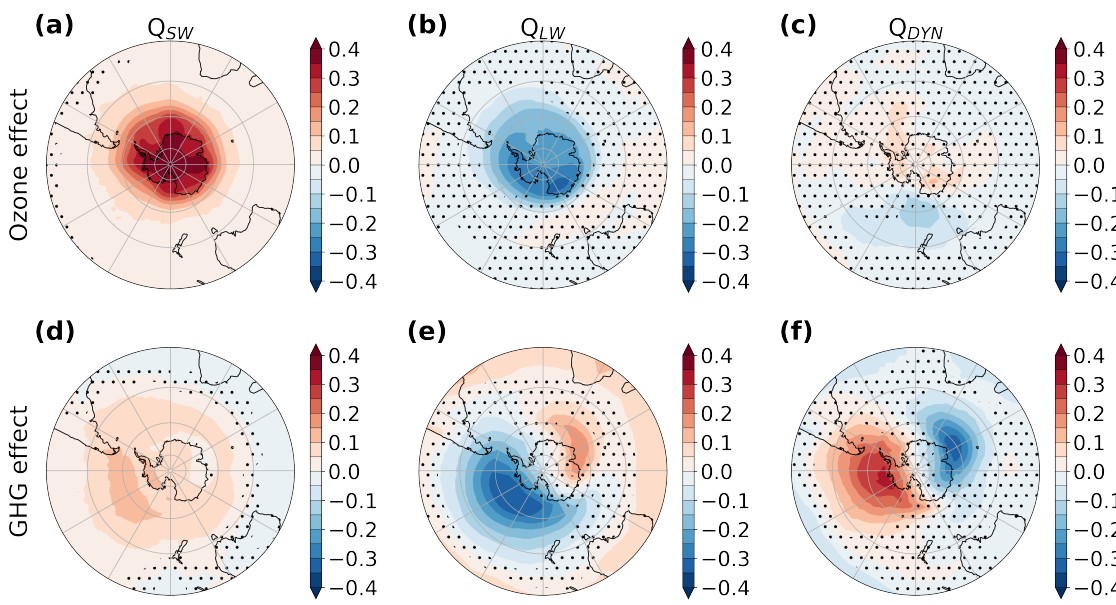

**Figure 6.** Changes in the SW (a and d), LW (b and e) and dynamical (c and f) heating rates (K day$^{-1}$) at 50 hPa during October due to ozone recovery (a-c) and GHGs (d-f). The stippling masks regions where the changes are not significant at the 95% confidence interval based on a two-tailed t-test.



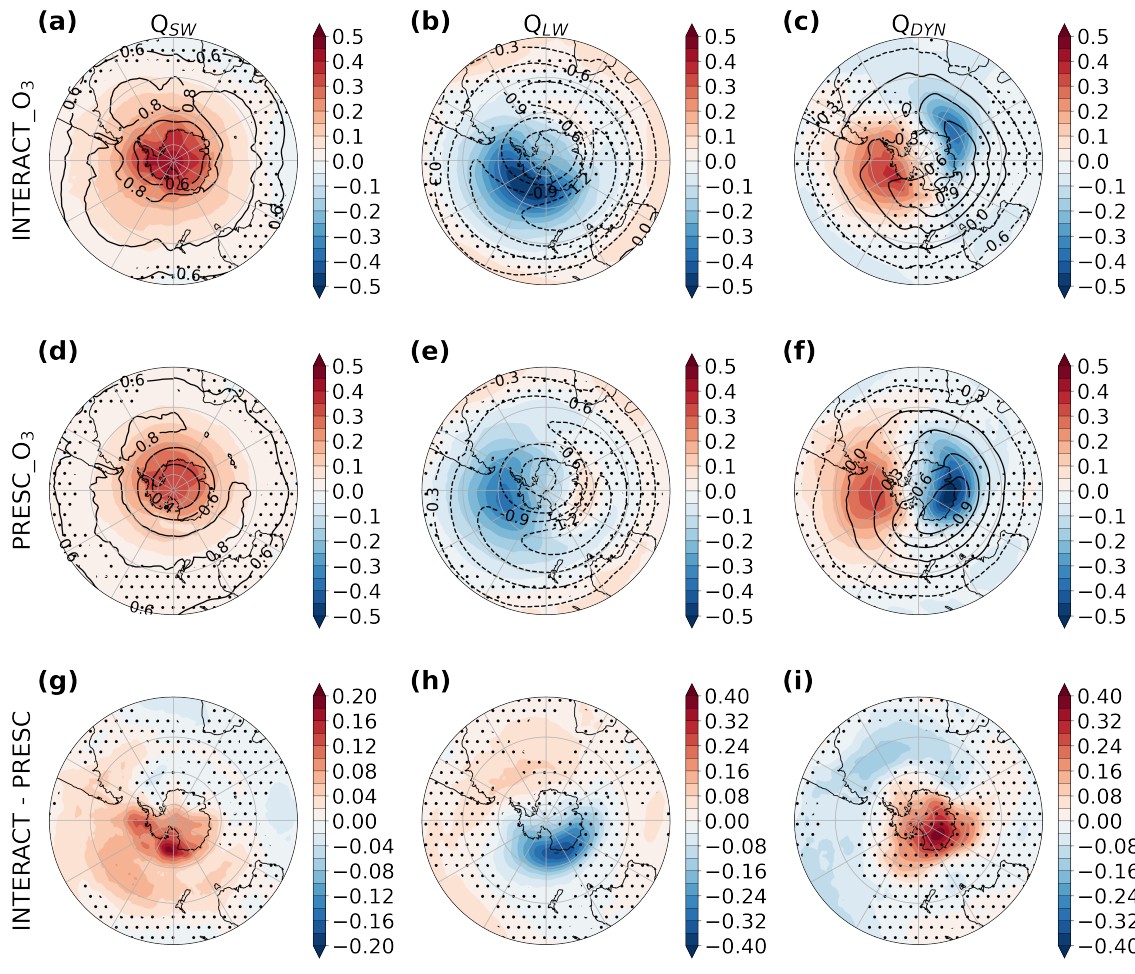

**Figure 7.** Changes in the SW (a and d), LW (b and e) and dynamical (c and f) heating rate (K day$^{-1}$) at 50 hPa during October due to ozone recovery and GHGs combined, in INTERACT_O$_3$ (a-c) and PRESC_O$_3$ (d-f), and the difference between the INTERACT_O$_3$ and the PRESC_O$_3$ changes (g-i). The stippling in a-f masks regions where the changes are not significant at the 95% confidence interval based on a two-tailed t-test, and the stippling in g-i masks regions where the difference is lower than $2\sigma$ inferred from the spread of the individual ensemble members. The contours in a-f show the current-day (2011-2030) climatologies in the respective fields.



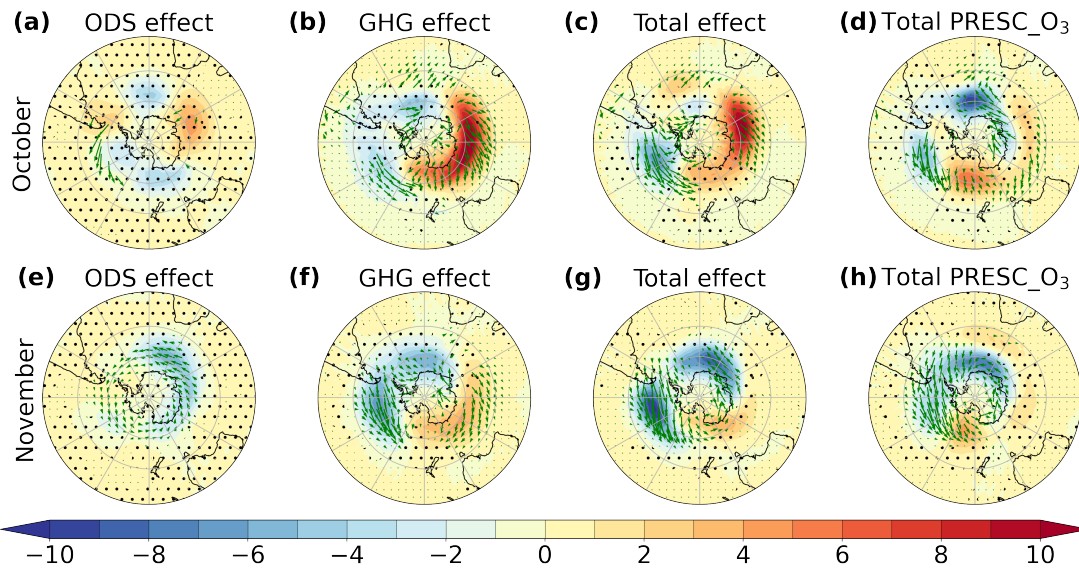

**Figure 8.** Changes in the flux of wave activity at 50 hPa during October (a-d) and November (e-h) due to ozone recovery (a and e), increasing GHGs (b and f) and both forcings in INTERACT_O$_3$ (c and g) and PRESC_O$_3$ (d and h). The color shading shows the change in the vertical ($10^{-3}$m$^2$ s$^{-2}$) and the vectors show the change in the horizontal (m$^2$ s$^{-2}$) component of the flux. The stippling masks regions where the changes are not significant at the 95% confidence interval based on a two-tailed t-test.

**Figure 9.** Changes in the SH downward mass flux (a-b), NH downward mass flux (c-d) and tropical upward mass flux (e-f) at 50 hPa for each month ($10^9$ kg s$^{-1}$). The changes due to ozone recovery are shown in blue, the changes due to GHGs are shown in orange (a, c, e), the total changes in INTERACT_O$_3$ are shown in black and the total changes in PRESC_O$_3$ are shown in gray (b, d, f). The error bars give the 95% confidence interval based on a two-tailed t-test. Differences between ensemble means are depicted by filled circles and differences between individual simulations are depicted by hollow circles.

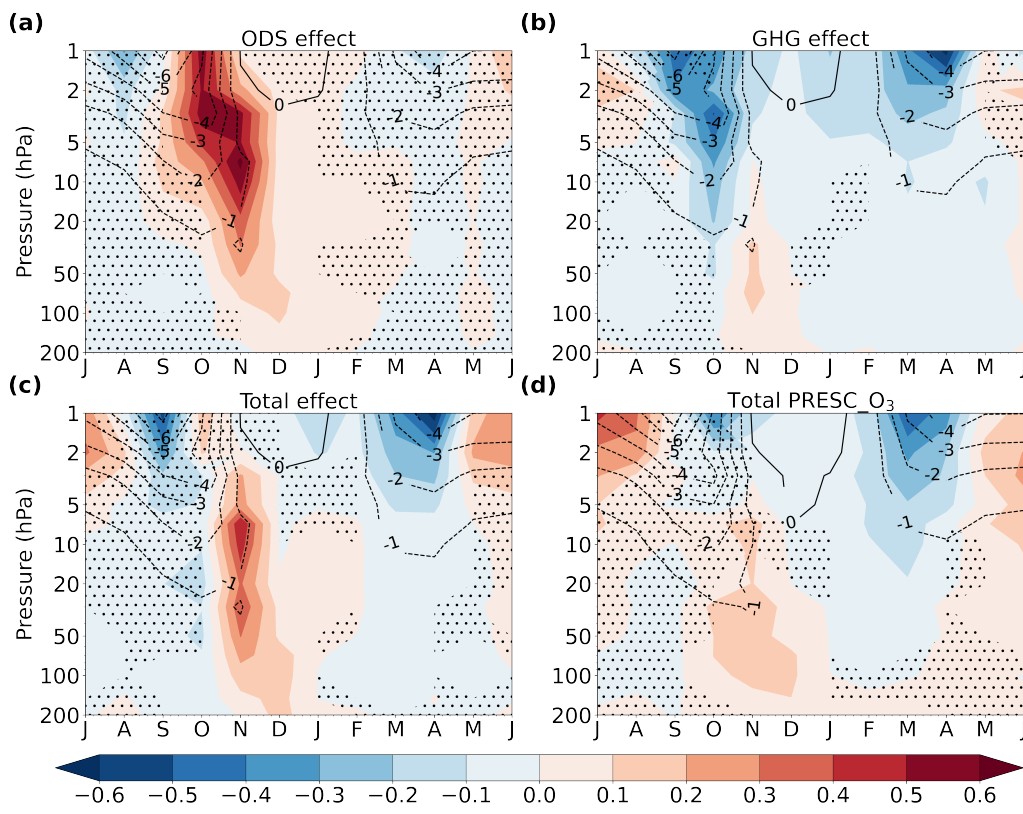

**Figure 10.** Changes in the polar cap (70°S-90°S) vertical residual velocity (in mm s$^{-1}$) for each month and pressure level (color shading): effect of ozone recovery (a), effect of GHGs (b), total effect in INTERACT_O$_3$ (c) and total effect in PRESC_O$_3$ (d). The contours depict the current day (2011-2030) climatology from INTERACT_O$_3$ in a-c and from PRESC_O$_3$ in d. The stippling masks regions where the changes are not significant at the 95% confidence interval based on a two-tailed t-test.

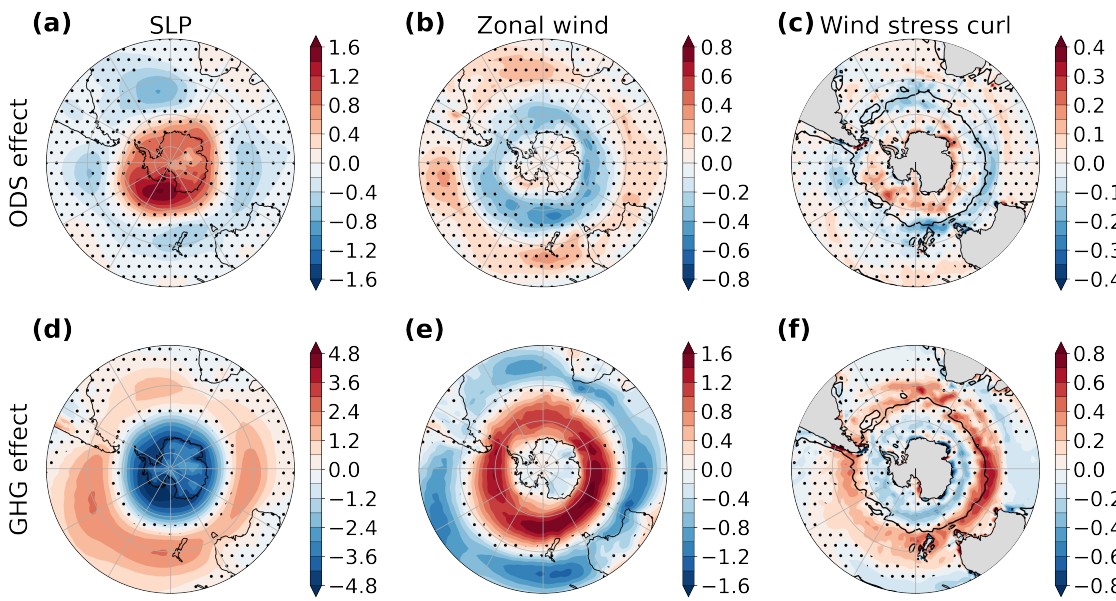

**Figure 11.** Changes in annual mean sea level pressure (a and d; hPa), surface zonal wind (b and e; m s$^{-1}$) and wind stress curl (c and f; $10^{-7}$ N m$^{-3}$) due to ozone recovery (a-c) and GHGs (d-f). The stippling masks regions where the changes are not significant at the 95% confidence interval based on a two-tailed t-test. The contour in c and f marks the location of zero wind stress curl.

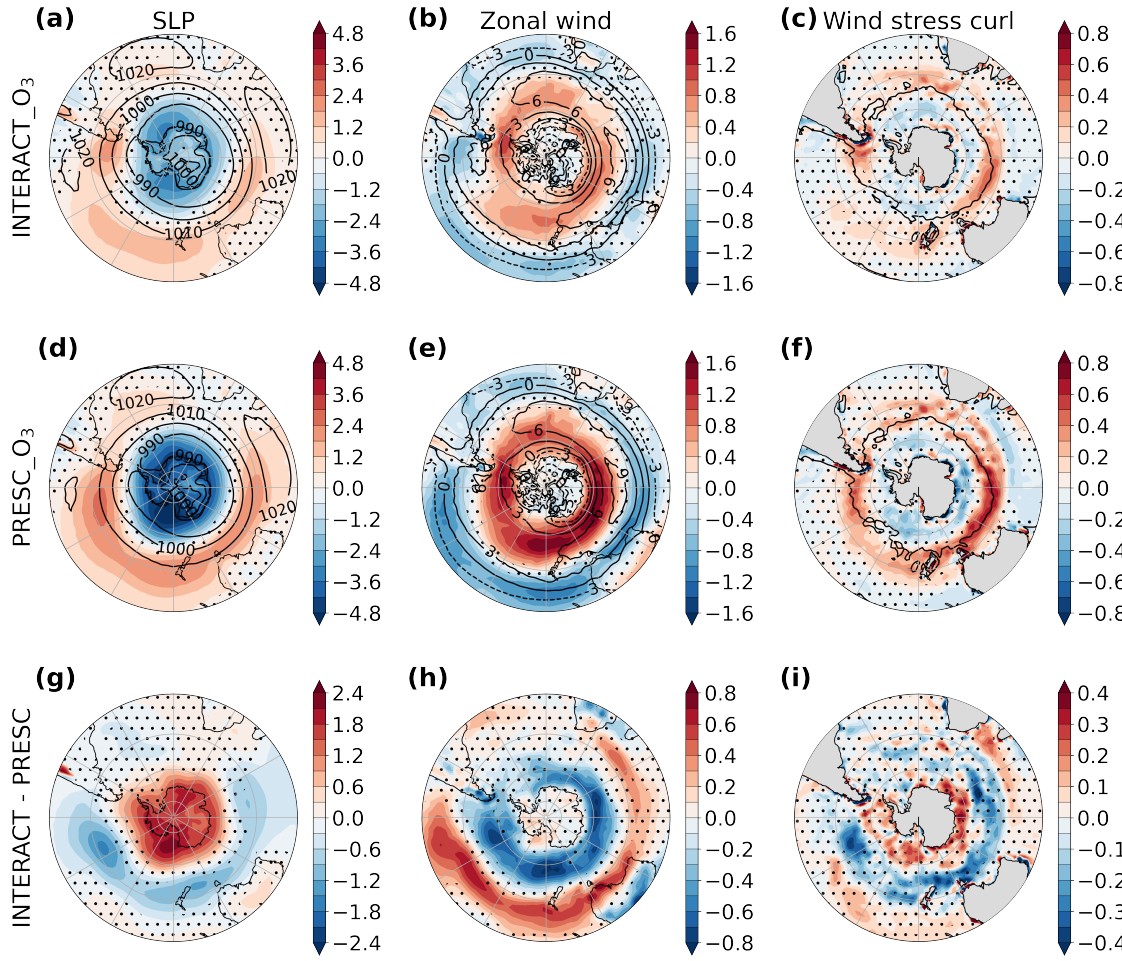

**Figure 12.** Changes in annual mean sea level pressure (a and d; hPa), surface zonal wind (b and e; m s$^{-1}$) and wind stress curl (c and f; $10^{-7}$ N m$^{-3}$) due to ozone recovery and GHGs combined, in INTERACT_O$_3$ (a-c) and PRESC_O$_3$ (d-f), and the difference between the INTERACT_O$_3$ and the PRESC_O$_3$ changes (g-i). The stippling in a-f masks regions where the changes are not significant at the 95% confidence interval based on a two-tailed t-test, and the stippling in g-i masks regions where the difference is lower than $2\sigma$ inferred from the spread of the individual ensemble members. The contour in c and f marks the location of zero wind stress curl, while contours in a, b, d and e show the current-day (2011-2030) climatologies in the respective fields.

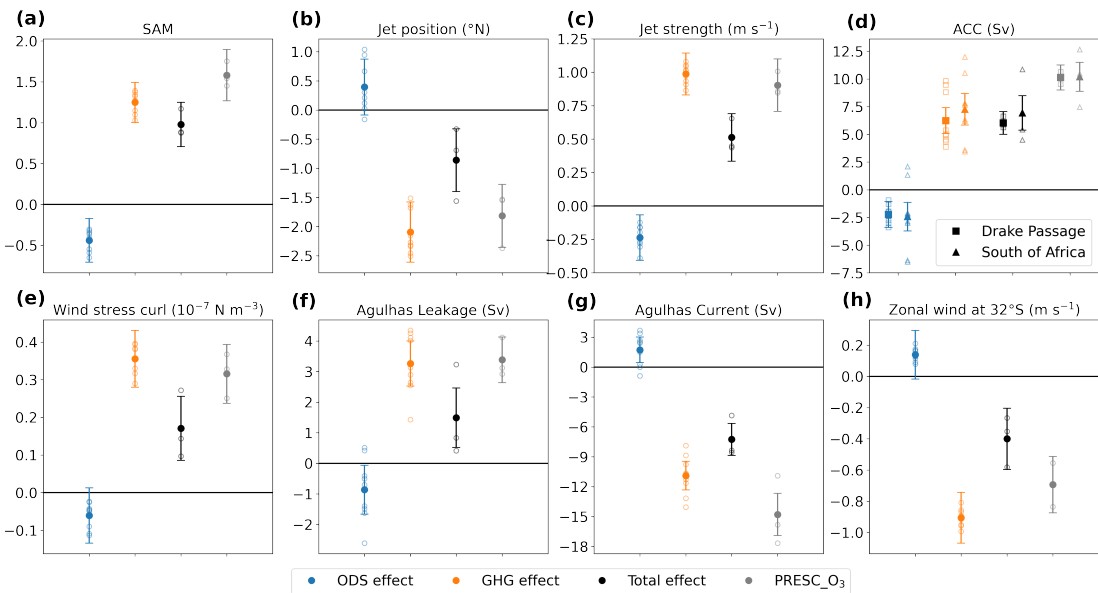

**Figure 13.** Changes in the annual mean SAM index calculated following the definition of Gong and Wang (1999) as the difference in the normalized zonal mean SLP between 40°S and 65°S (a), surface westerly jet position computed as the latitude of the maximum in the zonal mean zonal wind between 30°S and 65°S (b; ° of latitude, positive northward), surface westerlies strength averaged between 45°S and 60°S (c; m s$^{-1}$), ACC transport (d; Sv) in the Drake Passage, computed as the difference in the barotropic streamfunction between the Antarctic Peninsula and South America (circles), and south of Africa, computed as the maximum in the barotropic streamfunction between 20°E and 30°E (triangles), wind stress curl averaged between 40°S and 55°S and 30°E and 120°E (e; 10$^{-7}$N m$^{-3}$), Agulhas leakage transport (f; Sv), Agulhas Current transport (g; Sv) and the zonal wind at 32°S averaged between 30°E and 120°E (h; m s$^{-1}$). Filled symbols depict the change in the ensemble mean and hallow symbols depict the change in the individual members due to ozone recovery (blue) increasing GHGs (orange), combined forcing in INTERACT_O$_3$ (black) and combined forcing in PRESC_O$_3$ (gray). The error bars denote the 95% confidence interval based on a two-tailed t-test.

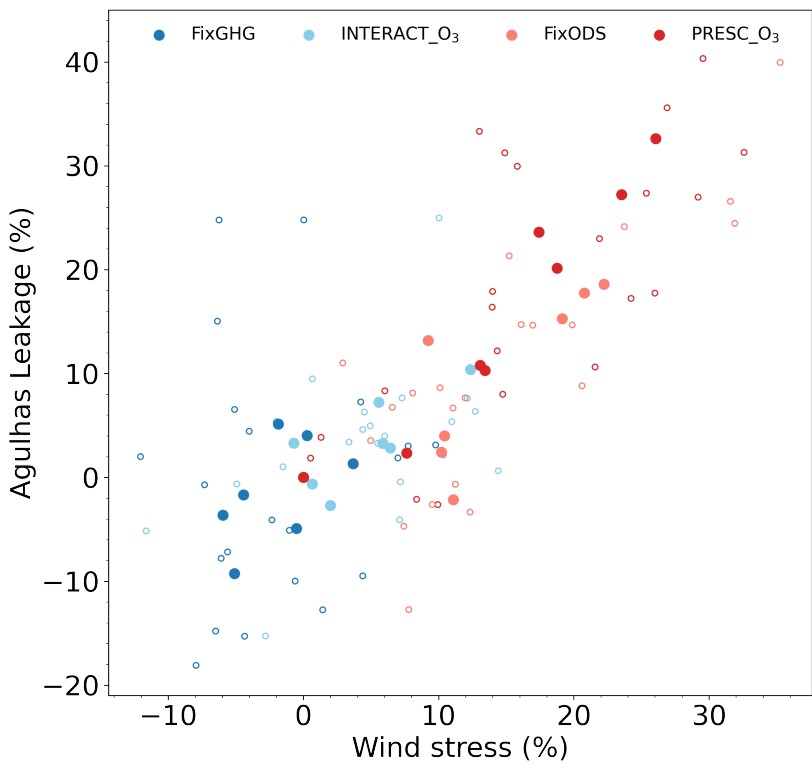

**Figure 14.** Decadal changes in Agulhas leakage versus decadal changes in wind stress over 45°S and 65°S and 30°E and 120°E (% change relative to 2014-2023). Each circle represents the change of each separate decade starting with 2014-2023 and ending with 2083-2094 relative to 2014-2023 in FixGHG (dark blue), INTERACT_O$_3$ (light blue), FixODS (light red) and PRESC_O$_3$ (dark red). Filled circles denote ensemble means and small, hallow circles denote individual members.



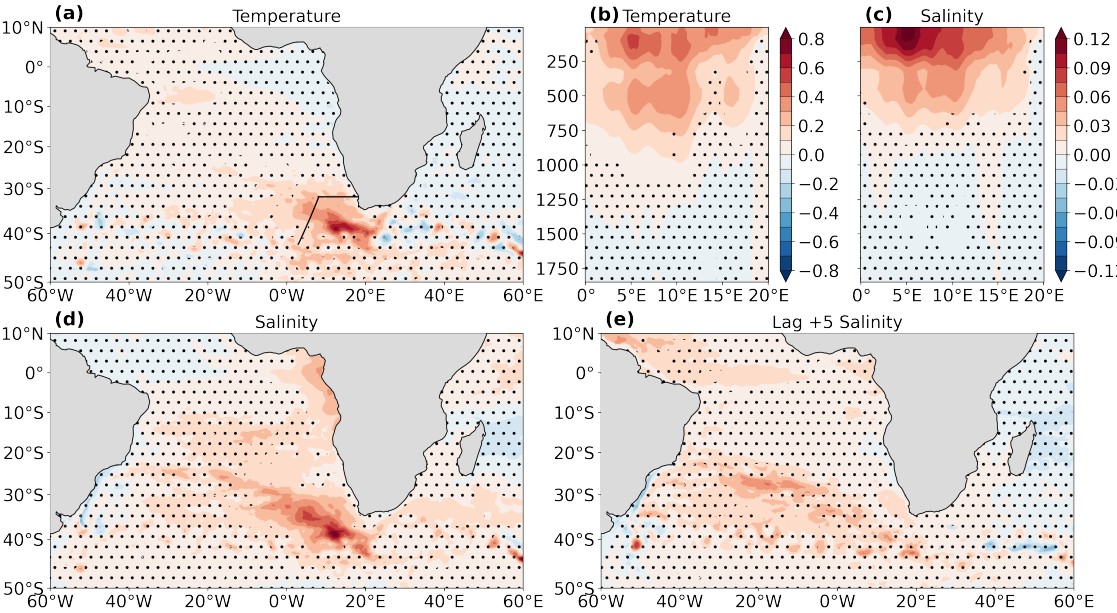

**Figure 15.** Composites of high Agulhas leakage periods for FixGHG: Anomalies with respect to the 2014-2094 mean of potential temperature (a; °C) and practical salinity at lag 0 years (d) and +5 years (e) averaged over the upper 1000 m of the ocean, and depth profiles of potential temperature (b) and practical salinity (c) anomalies between 0° and 20°E at 35° S. The fields are low-pass filtered to retain only variations with a period above 5 years and detrended by removing a linear trend. The stippling masks anomalies that are not significantly different from the time mean according to the Monte Carlo method (see text for details). The line in panel a marks the Good Hope section used in calculating the Agulhas leakage.



**Table 1.** Separation of the different effects investigated in this study. The first column gives the effect that is extracted, the second column lists the ensembles that are subtracted to extract an effect, and the third column gives the periods over which the differences are taken.

| Effect | Simulations | Period |
|---|---|---|
| ODS | INTERACT_$O_3$ - FixODS | 2075-2099 |
| GHG | INTERACT_$O_3$ - FixGHG | 2075-2099 |
| Total interactive chemistry | 2090 INTERACT_$O_3$ - 2020 INTERACT_$O_3$ | 2080-2099 vs. 2011-2030 |
| Total prescribed chemistry | 2090 PRESC_$O_3$ - 2020 PRESC_$O_3$ | 2080-2099 vs. 2011-2030 |