# Peer review of "Figure S1. Ozone recovery in FOCI: time series of polar cap ( $60^{\circ}\text{S}$ - $90^{\circ}\text{S}$ ) total column ozone for October (DU). The INTERACT\_O3 ensemble mean is shown in orange and the individual members in gray. The horizontal green lines mark the 1980 and 1960 ozon"

_Weather and Climate Dynamics, 2021_

## Author Comment (AC1)

Dear Referees, Dear Editor,

Thank you for the detailed review of our manuscript and for the valuable criticism and suggestions, which helped us to improve the manuscript significantly. We first summarize the major changes to the manuscript:

- We condensed the manuscript by eliminating non-essential information from the abstract and introduction and by merging the sections on wave activity and the Brewer-Dobson circulation and the discussion and summary sections, thereby removing repetitive statements.
- We included figures showing the signature of the planetary zonal wavenumber (PW) 1 changes in the temperature field (Fig. 7) and in the divergence of the EP flux (supplementary Fig. S3) and we took the changes in PW 1 into consideration in the interpretation of the stratospheric changes induced by increasing greenhouse gases (GHGs).
- We extended the analysis of the oceanic changes to include sea surface temperature and depth-latitude sections of temperature. In order to make space for these new figures, we moved former Fig. 14, showing the scatter of Agulhas leakage versus wind stress, to the supplement.
- We rephrased where necessary to clarify that the differences between the ensemble with interactive ozone chemistry and the ensemble to which the CMIP6 ozone field was prescribed arise primarily due to the weaker ozone recovery exhibited by the CMIP6 ozone field.

Please find below our point-by-point answers to your comments. The line numbers we give refer to the version of the manuscript with tracked changes.

**Anonymous Referee #1**

This manuscript examines the impact of ozone recovery and increases in greenhouse gases over the 21st on the Southern Hemisphere, including changes from the stratosphere to the oceans, using ensembles of chemistry-climate model simulations with differing ozone and GHG fields. The manuscript contains several interesting results that are suitable for publication in Weather and Climate Dynamics. However, before publication I think the presentation needs to be more concise (and in one aspect more precise), and the analysis of the ocean changes needs to be broadened.

**Major Comments**

The paper is very long, and needs to be more concise. This is especially the case for the Abstract and the Introduction. For example, it is not until page 5 of the Introduction that what is actually

done in the paper is mentioned. I suspect you will lose readers before they get to this stage. While the review is good, much of it is not needed to justify the analysis done.

We condensed the introduction by eliminating information that was not essential for our analysis. The abstract is now also more concise.

Also the discussion section repeats much of what is in the results section, and I think could easily be removed (with maybe a few sentences comparing to previous studied included in the results section).

We removed the discussion section and moved important statements either to the summary section, which is now titled "Summary and discussion", or to the results section.

2. Some of the text discussing the differences of the INTERACT_O3 and PRESC_O3 runs is misleading, or at least could easily be misinterpreted. For example in both the abstract (line 9-10) and Conclusions (line 1049) there are statements about significance differences between the INTERACT_O3 and PRESC_O3 simulations that will read as if these differences are due whether the ozone field is prescribed or calculated interactively.

While interactive vrs prescribed may be causing some differences, no evidence is provided to show this is the major cause. Given the differences in the ozone fields shown in Fig 1, I actually think this the major cause of the differences between runs and not interactive vrs prescribed ozone. This is mentioned in section 5.7 and the conclusions, but more focus is put on the prescribed versus interactive aspect in these discussions, and in other places the writing is such to strongly indicate it is due to ozone not being interactive in the prescribed ozone run.

Note, I don't understand the statement on line 981 "the fact that the differences in some fields are as large as the impact of ozone recovery, it is likely not the only cause".

I don't disagree that fact ozone is no interactive in prescribed run will likely be making a cause, but to make strong statements on this you would need to do PRESC_O3 runs that use ozone from the INTERACT_O3 (as has been done in many previous statements). Until you do this I don't think you can make statements on the importance of the ozone being interactive.

We agree that the differences in the various fields between the ensemble with interactive chemistry and the ensemble with prescribed CMIP6 chemistry arise predominantly from the differences in the FOCI and the CMIP6 ozone fields. We made changes throughout the manuscript to reflect this more clearly, such as renaming the PRESC_$O_3$ ensemble to CMIP6_$O_3$, always mentioning that the prescribed ozone field is that for CMIP6 and placing more emphasis on the difference between the FOCI and the CMIP6 ozone fields.

The statement in the abstract previously found at lines 9-10 was removed in the process of making condensing the abstract. The statement regarding the differences between INTERACT_$O_3$ and CMIP6_$O_3$ in the abstract now reads:

"This second ensemble simulates a weaker ozone effect in all the examined fields, consistent with its weaker increase in ozone. The magnitude of the difference between the simulated changes at the surface and in the oceanic circulation in the two ensembles is as large as the ozone effect itself. This shows the large uncertainty that is associated with the choice of the ozone field and how the ozone is treated." (lines 34-37)

The statement at former line 1049 was also replaced in the process of merging the summary and the discussion.

We rephrased the statement that was previously found at line 981 (now found in the third to last paragraph of the manuscript at lines 1342-1343) to express the point more clearly. We are trying to explain that the magnitude of the differences between INTERACT_$O_3$ and CMIP6_$O_3$ in the surface and oceanic circulation changes hints to the fact that the weaker ozone increase in CMIP6 cannot explain these differences entirely (although it probably explains a large part). More concretely, let us take the example of the change in the surface westerly wind (Fig. 15c). The difference between the change in INTERACT_$O_3$ and CMIP6_$O_3$ is about 0.3 ms$^{-1}$. The ozone effect has a magnitude of about 0.25 ms$^{-1}$. Therefore, the difference between INTERACT_$O_3$ and CMIP6_$O_3$ is larger than the ozone effect itself. Since we do not expect large differences in the GHG effect between the two ensembles, this would imply that there is no ozone recovery at all in CMIP6_$O_3$, which is clearly not the case. Since CMIP6_$O_3$ exhibits ozone recovery (albeit weaker than INTERACT_$O_3$), other factors must be additionally responsible for these difference. These factors are discussed in the (now combined) "Summary and discussion" section: interactive vs prescribed ozone, prescribing monthly rather than daily ozone and spatial inconsistencies between the prescribed ozone field and the simulated dynamics. Since this is the discussion section, we find it appropriate to rely on the findings of past studies to offer additional explanations for our results, even if we did not prove these explanations in the current manuscript.

3. I find the choice of ocean diagnostics a little puzzling. I know that the Agulhas leakage is an important aspect of the ocean circulation, but I am not sure it is the most important aspect to focus on and I think before focusing on a regional aspect (e.g., Agulhas leakage) I think there should be some discussion of these more hemispheric aspects.

In fact there have several recent studies that have shown the impact of ozone depletion on the meridional overturning circulation, sea surface and interior temperatures, and sea-ice (e.g. Li et al 2016, Ferreira et a 2015, Seviour et al 2016, 2019). These aspects and studies should be discussed in the Introduction, and some analysis of these fields should be included. Doing this would not only present a better view of the ocean changes, but would also enable comparison with these studies (esp the Li et al 2016 and Seviour et al 2016 studies which also examined aspects of the impact of using prescribed ozone). For example, are the differences in these aspects between INTERACT_O3 and PRESC_O3 consistent with these studies?

Our model setup with the INALT10X high resolution nest in the South Atlantic and western Indian Ocean is a valuable tool for studying the circulation changes in this region, as mesoscale features

such as the Agulhas rings and eddies are resolved. As the model does not have the same advantage for the rest of the global ocean, we focus our analysis on this region, which on its own offers plenty of interesting changes. Changes in Agulhas leakage in the past have been linked to the formation of the ozone hole, but the future impacts of ozone recovery and increasing GHGs on the Agulhas leakage have not been studied before. As such, our analysis of the Agulhas leakage adds new and interesting insights to our current knowledge about the Agulhas leakage and watermass changes in the South Atlantic. Our model setup is fitting for this scope. We additionally investigate the response of the ACC, measured in the Drake Passage and South of Africa, to ozone recovery and GHGs. The analysis of the ACC also benefits from the high resolution nest, as changes in eddy activity are known to be important for the response of the ACC to increased wind stress forcing (e.g., Farneti et al., 2010; Munday et al., 2013).

We extended our analysis, as suggested, to also include impacts on ocean temperature, focusing again on the nest region (Figs. 17 and 18) and we discussed these changes in relation to the suggested references, as well as to other publications (e.g. at lines 979-983). To explain the simulated temperature changes, we additionally added the changes in the subtropical gyre and in the Ekman transport and pumping in the supplement. These additions considerably extended our already very comprehensive manuscript.

Further adding other aspects, such as changes in the (A)MOC would lengthen the paper by too much. Changes in the AMOC are also, to an important extent, driven from the North Atlantic (as can be seen in Fig. R1) and disentangling the northern vs. the southern influence could make the topic of a publication on its own. Furthermore, the studies by Li et al. (2016), Ferreira et al. (2015) and Seviour et al. (2016) only showed MOC changes in depth coordinates and as such they simply report a shift/strengthening of the Deacon cell. The Deacon cell is an artificial cell present only when the MOC is viewed in depth coordinates and it vanishes when density coordinates are used (e.g., Döös and Webb, 1993). Additionally, due to the eddy compensation effect, a realistic simulation of the MOC would require high enough resolution over the entire Southern Ocean, not just over parts of it. Also important is certainly the fact that our high-resolution nest is limited to 63°S. Processes such as bottom water formation, also impacting the AMOC, are simulated at different, much coarser resolution.

We therefore consider the choice of oceanic fields appropriate and we added a statement in the introduction to make it clear from the beginning that our focus will be on the South Atlantic.

"We systematically separate the effects of the two drivers starting in the SH stratosphere and ending with the circulation in the South Atlantic Ocean and its corresponding Southern Ocean sector." (lines 167-169)

Regarding the comparison between the total changes in the ensemble with interactive ozone (INTERACT_$O_3$) and the ensemble to which the CMIP6 ozone field is prescribed (CMIP6_$O_3$), we find that the ozone effect on the surface winds and the ocean is weaker in the latter ensemble in agreement with the findings of Li et al. (2016) and Seviour et al. (2016). However, our results are not directly comparable with those two studies. The differences between the simulations with

interactive and prescribed ozone in the study by Li et al. (2016) arise primarily due to the fact that zonal mean ozone is prescribed and therefore ozone asymmetries are not included. The CMIP6 ozone we prescribe is three-dimensional and includes ozone asymmetries. While prescribing monthly instead of daily ozone likely plays a role in the differences we see between INTERACT_O$_3$ and CMIP6_O$_3$, in agreement with Seviour et al. (2016), it is only a secondary role. The differences in our study are primarily explained by the differences in the ozone fields and their different rate of recovery.

Döös, K., & Webb, D. J. (1994). The Deacon Cell and the Other Meridional Cells of the Southern Ocean, *Journal of Physical Oceanography*, *24*(2), 429-442.

[Figure]

Figure R1: Changes in the AMOC (Sv) due to ozone recovery (a), increasing GHGs (b) and their combined effect in INTERACT_O$_3$ (c) and CMIP6_O$_3$ (d). The stippling masks regions where the changes are not significant at the 95% confidence interval according to a two-tailed t-test.

**Minor Comments:**

Line 4: "... a unique coupled ..." Why "unique". I guess at one level all models are unique, but I don't see what is specifically unique about model used here.

To our knowledge, the configuration of FOCI discussed here is currently the only configuration of a coupled climate model that combines both the ability to calculate the ozone chemistry interactively and the high resolution ocean, thus making it unique. Coupled ocean-atmosphere chemistry-climate models typically do not have a high enough ocean resolution to resolve mesoscale features, while the few coupled climate models that have a high resolution ocean do not include interactive chemistry in the atmosphere. For example, Cheng et al. (2018) used a climate model with 0.1 ocean resolution to study Agulhas leakage variability, but their model does not include interactive ozone chemistry. In contrast, the comprehensive two part study by McLandress et al. (2010, 2011) used a coupled chemistry climate model to separate the impacts of ozone and GHG changes on the atmospheric circulation, but the ocean resolution in their model is only 1.86°.

Line 300: I think the evolution of TCO (Fig S1) should be shown in the main text, and as well as showing INTERACT_O3 the evolution of other runs should be shown. As well as interpreting differences between runs shown here this will have put the ozone evolution in context of other CCMs studies.

We moved former Fig. S1 to the main text, where it is now Fig. 2 and we added the October Antarctic TCO evolution for the FixODS and FixGHG ensembles.

Line 310-314: It is stated that the ozone fields show a similar vertical and seasonal structure of ozone recovery, but when I look at fig 1 I see big differences in magnitude and vertical structure in SON months, and in the following sentences some of these large differences are discussed. So I think it is not correct to same similar unless you describe what is meant by this.

We rephrased to clearly point out the similarities between the two fields: "Both fields show the recovery of the stratospheric ozone, exhibiting a maximum increase in ozone in the upper stratosphere and, during austral spring, between 100 hPa and 20 hPa." (lines 358-359)

**Anonymous Referee #2**

The study by Ivanciu et al investigates the effects of future changes in both ODS (leading to ozone recovery) and GHG concentrations on the stratospheric circulation, and consequent effects on surface winds and oceanic circulation. The study adds to the existing literature on the individual and combined effects of ODS and GHG changes, mostly confirming past results on stratospheric(-tropospheric) circulation changes. A new component of the work by Ivanciu et al is the use of a coupled ocean-atmosphere model with a relatively well resolved ocean, including nesting of some regions to eddy-resolving resolution. The paper includes the investigation of southern hemispheric ocean circulation changes induced by ODS/ozone and GHG changes. Thus, the study presents analysis of circulation changes spanning from the stratosphere to the ocean. The results on the stratospheric circulation are presented in detail, and given that they mostly confirm past

work the section could be condensed at places (see general comment below). The authors claim to have found a new result in that the GHG-induced temperature/zonal wind changes bear hemispheric asymmetry. However, the physical interpretation of the asymmetry given in the paper is in my opinion misleading (see major comment). Therefore, I overall can recommend the paper for publication in WCD only after major revision. Note that my review focuses mostly on the stratospheric/atmospheric part of the analysis in the paper, as the oceanic analysis is beyond my expertise to judge.

**Major comment**

The authors stress at multiple places in the study the result of hemispheric asymmetric GHG-induced changes in temperature, winds and wave activity, and claim for example that "GHGs bring an important contribution to the total spring temperature changes at this height. This contribution is largely underestimated if zonally-averaged fields are investigated (Fig. 2b), as the cooling pole offsets a large part of the warming pole" (line 367-369).

This statement seems to imply that the GHGs exhibit an asymmetric forcing, that is missed if the zonal average is considered, seemingly implying that there are distinct different physical processes at work over the eastern versus western hemisphere. As the authors do state later in the paper, the asymmetries are related to dynamical processes, namely planetary wave activity. Clearly, the asymmetric features the authors refer to are the signature of a planetary wave (of wavenumber 1). As such, the asymmetric response is simply the result of changes in PW activity, and thus one single hemispheric phenomenon and not distinct processes in the two hemispheres. The PW signature noted in the paper is an interesting result, but it should be interpreted clearly as a change in planetary wave amplitude (and possibly phase) that is brought upon by the GHG-induced changes to the circulation and likely non-linear changes of the propagation conditions of PW 1, rather than implying that the hemispheric asymmetric response is an inherent response to the GHG-forcing directly.

We agree with the fact that the stratospheric response (temperature, zonal wind, residual circulation) to increasing GHGs arises due to dynamical processes, specifically due to changes in the planetary wavenumber (PW) 1. We wish to clarify that we do not claim, nor did we wish to imply, that GHGs exhibit an asymmetric forcing, or that the spring lower stratospheric temperature changes that occur in response to GHGs are simply a radiative response (as it is, for the most part, the case for ozone).

In fact, at lines 420-423 in the original manuscript, we stated: "This pattern is driven by the dynamical response to increasing GHGs (Fig. 6f). The dynamical heating rate change due to GHGs explains the temperature response to GHGs almost entirely, with an additional weak contribution due to the increase in the SW heating rate over the Pacific sector caused by the dynamical ozone increase in this region." It appears, however, that our phrasing in some other parts of the manuscript gave this wrong impression and we modified the statements where we thought this might be case. We additionally now refer to the changes as changes in the characteristics of the PW1.

"We will show below that this is caused by a change in the PW1 in response to GHGs leading to contrasting dynamical heating responses to GHGs over different sectors of the Southern Ocean that compensate each other in the zonal mean." (lines 379-381)

"Comparing the current day structure with the changes caused by GHGs (Fig. 7b), it becomes clear that the increase in GHGs leads to an eastward phase shift of the PW1." (lines 419-420)

We consider the information formerly found at lines 367-369 correct. As we can see from Fig. 5e, the temperature changes that result from dynamical changes associated with the increase in GHGs have a magnitude comparable to those attributed to ozone recovery (Fig. 5b). In addition, looking only at zonal mean temperature changes in response to increasing GHGs (Fig. 3b), one would conclude that there are no significant changes in response to GHGs occurring during austral spring between approximately 200 hPa and 20 hPa. The map in Fig. 5e, clearly shows that increasing GHGs lead to significant temperature changes, however. We rephrased to make it clear that the temperature changes we refer to in response to GHGs are not a direct radiative effect, but arise due to dynamical changes:

"This indicates that the increase in GHGs results in important temperature changes at this height in the lower stratosphere during spring, substantially contributing to the total temperature change in Fig. 6b. However, although the temperature changes resulting from ozone recovery and increasing GHGs have similar magnitudes, the mechanisms behind these changes are different. As shown above, while the temperature changes caused by ozone recovery (Fig. 5b) result primarily from radiative processes, the changes occurring in the springtime lower stratosphere due to GHGs (Fig. 5e) are a result of dynamical processes. The GHG-induced springtime lower stratospheric temperature changes are largely underestimated if zonally-averaged fields are investigated (Fig. 3b), as the PW1 anomalies cancel out to a large degree." (lines 432-440)

The well established (linear) theory of (stratospheric) general circulation dynamics decomposes the atmospheric flow into a zonal mean component and deviations from this flow, i.e. atmospheric waves, that propagate on the zonal mean "background". The interaction of the planetary waves and the background zonal mean flow is the dominating  process that forms the stratospheric circulation. Employing this framework, the results can be interpreted more easily and related to the known mechanism of stratospheric dynamics: The temperature anomalies suggest a reduction of PW amplitude and slight phase change (see Fig. 5b / e). This is consistent with reduced vertical wave propagation (as measured by the eddy heat flux, Fig. S5) in Oct/Nov, and consequently reduced EP flux divergence (see Fig. S6), driving a reduction in the residual circulation below (Fig. 10), consistent with the well known "downward controlled" wave driving of the (steady state) residual circulation. How the reduction in PW activity comes about is another interesting research question, that probably would involve more in-depth analysis and might not be straight forward to answer due to the highly non-linear nature of wave - mean-flow interactions.

The authors employ the 3-D Plumb flux to investigate the changes in wave activity, and find again opposing signals in the hemispheres - which again is consistent with a PW 1 signature. The 3-d wave activity flux is a useful tool to study local effects of wave propagation of smaller (higher wavenumber) waves, but a planetary wave is by definition a planetary phenomenon, and therefore the 3-d wave activity flux is in my opinion not suitable to study planetary wave propagation. Further, it cannot be easily used to explain the changes in the residual circulation, opposed to the conventional EP flux divergence.

We, use the eddy heat flux and EP flux divergence to explain the springtime changes in the residual circulation driven by both the increase in GHGs and ozone recovery, according to the "downward control" principle. In addition, we extend the analysis by investigating the spatial patterns of the changes. The EP flux is by definitional a zonal mean field, therefore we additionally employ the 3D Plumb flux. The Plumb flux has been used to investigate planetary wave propagation in a number of previous studies (e.g., Kodera et al., 2008, 2013; Lubis et al., 2016; Nath et al., 2016; Harada and Hirooka, 2017). In the zonal mean, its vertical and meridional components are equivalent to the components of the EP flux. Indeed, we find very good agreement between the changes in the Plumb flux and the changes in the eddy heat flux (equivalent to the vertical EP flux component) and the EP flux divergence. We therefore consider investigating the spatial patterns given by the Plumb flux useful and we show the changes in the Plumb flux in the supplement.

Kodera, K., Mukougawa, H., and Itoh, S. (2008), Tropospheric impact of reflected planetary waves from the stratosphere, Geophys. Res. Lett., 35, L16806, doi:10.1029/2008GL034575.

Kodera, K., Mukougawa, H., and Fujii, A. (2013), Influence of the vertical and zonal propagation of stratospheric planetary waves on tropospheric blockings, J. Geophys. Res. Atmos., 118, 8333–8345, doi:10.1002/jgrd.50650.

Lubis, S. W., Matthes, K., Omrani, N., Harnik, N., & Wahl, S. (2016). Influence of the Quasi-Biennial Oscillation and Sea Surface Temperature Variability on Downward Wave Coupling in the Northern Hemisphere, *Journal of the Atmospheric Sciences*, *73*(5), 1943-1965.

Nath, D., Chen, W., Zelin, C. *et al.* Dynamics of 2013 Sudden Stratospheric Warming event and its impact on cold weather over Eurasia: Role of planetary wave reflection. *Sci Rep* **6,** 24174 (2016). https://doi.org/10.1038/srep24174

Harada, Y., & Hirooka, T. (2017). Extraordinary features of the planetary wave propagation during the boreal winter 2013/2014 and the zonal wave number two predominance. Journal of Geophysical Research: Atmospheres, 122, 11,374– 11,387. https://doi.org/10.1002/2017JD027053

Therefore, I recommend that the authors revise the interpretation of their results throughout the paper, namely interpreting the asymmetric response as a change in planetary wave activity. Employing a wavenumber decomposition of the fields and calculating the wave (EP) fluxes for

the individual wavenumbers would help to build a consistent interpretation of the simulated changes in the stratospheric circulation.

We included the PW1 temperature changes in Fig. 7 and the changes in the EP flux divergence associated with the PW1 in supplementary Fig. S3, while the total EP flux change, now with changes in zonal wind overlaid, was moved in the main manuscript (Fig. 12). We rephrased our statements throughout the manuscript to refer to the asymmetric changes as a change in PW1.

This extends to the discussion, where it is stated that "contrasting results can be explained by intermodel differences in simulating the strength of the GHG-induced change within each pole, which lead to different levels of compensation between the warming and the cooling pole."

I agree in that there are (likely) large intermodel differences, but the interpretation here would rather be that different models simulate the changes in planetary wave activity differently. This is again likely a result of the highly non-linear nature of wave - mean-flow interaction, so that for example slight differences in the model basic state might lead to a difference in the response of planetary wave / polar vortex dynamics. This might be somewhat analogous to the northern hemisphere, where models simulate a large range of different responses of the polar vortex to increasing GHG concentrations, and the reasons for this are still not entirely clear (e.g., Wu et al, 2019, GRL).

We rephrased to "These contrasting results are likely explained by intermodel differences in simulating the changes in planetary wave activity due to GHGs, which result in temperature and zonal wind responses of different magnitudes in the different models." Please note that the statement is now found in the "Summary and discussion" section at lines 1204-1206. This is a consequence of the fact that we merged the discussion and summary sections in order to condense the manuscript, as suggested.

**Minor comments**

General comments:
1. The authors compare one set of all-forcing simulations with interactive chemistry to a simulation in which the ozone field provided for CMIP6 is prescribed. This is a fair and useful comparison (as the CMIP6 ozone is used by many models), but the comparison of the simulations mixes two effects: 1) interactive versus prescribed ozone and 2) using an entirely different ozone field (model's own ozone versus CMIP6 ozone). Those different effects are acknowledged at a couple of places in the paper (e.g., line 414), but I would recommend to clarify this from the beginning (e.g. line 173; line 229) to not mislead the reader into thinking this comparison would quantify the effects of the interactive nature of ozone alone. The authors might even want to consider to rename the "prescribed O3" simulation into "CMIP6-O3" to avoid this misinterpretation. Apart from the interactive nature of ozone in one set of simulations (which is certainly more realistic than prescribed ozone in general), it is hard to judge which ozone field will represent future ozone changes more realistically - so the comparison does rather tell us about the possible uncertainty in climate projections arising from a different treatment of ozone.

The authors might want to consider to rephrase a few sentences to acknowledge this fact more clearly (e.g., line 964:: "GHG effect is more dominant when the ozone field is prescribed." - this likely depends on the exact ozone field that is prescribed, rather than on whether it is prescribed or calculated interactively).

We renamed the PRESC_$O_3$ ensemble to CMIP6_$O_3$, as suggested, and we made changes throughout the manuscript to put more emphasis on the role played by the differences between the CMIP6 ozone field and the ozone field simulated in FOCI.

The statement that was previously found at line 173 now reads: "We compare our future SH climate projections from the interactive chemistry version of the model with projections from the same model, but in the configuration that prescribes the CMIP6 ozone field. This comparison is relevant as not all the models participating in the current CMIP phase include interactive chemistry and therefore have to prescribe this ozone field. Prescribing the ozone field has been shown to alter the response of the SH circulation to ozone changes because of the temporal interpolation from monthly mean values to the model's time step (Neely et al., 2014; Seviour et al., 2016), and the missing ozone asymmetries (Waugh et al., 2009b; Li et al., 2016), and ozone-radiative-dynamical feedbacks (Haase et al., 2020) and spatial inconsistencies between the prescribed ozone and the polar vortex (Ivanciu et al., 2021). The CMIP6 ozone field includes zonal variations, but the rest of the above-mentioned issues, together with a different magnitude of ozone changes than simulated by the model, are expected to lead to differences in the effects of ozone recovery between the version of the model with interactive ozone and with prescribed CMIP6 ozone." (lines 184- 200)

The statement previously found at line 225 (we assume you meant 225 rather than 229) now reads "This total effect is then compared to that obtained from INTERACT_$O_3$ to investigate the dependency of our results on the ozone field used." (lines 269-270)

The statement previously found at line 964 was changed to "The comparison of the combined ozone and GHG effect between the ensemble with interactive ozone chemistry, INTERACT_$O_3$ , and the ensemble in which the CMIP6 ozone field is prescribed, CMIP6_$O_3$, revealed that the ozone effect is weaker and the GHG effect is more dominant in the latter ensemble, in agreement with the weaker increase in the ozone field in CMIP6." (lines 1338-1340)

2. The paper presents extensive analysis and diagnostics, and is as such very long. The authors might want to consider to condense the material at a few places (in particular Sec. 4.1.1-4.1.3; and in Introduction, Discussion/Summary, see specific comments below). It might help to shift the focus on the new results, e.g. on the (mechanism for) GHG-induced PW attenuation and weakening of circulation in spring in Section 4.1.

We shortened the introduction by removing nonessential information, we merged sections 4.1.2 and 4.1.3 and we removed the discussion section, moving the important content to the summary section, which is now called "Summary and discussion", or to the results section.

Specific comments

- line 54: to my knowledge the evidence of effects of ozone depletion on carbon uptake are very weak and the link is not established (see e.g. WMO O3 assessment, Chapter 5, 2018).

We removed the statement in the process of shortening the introduction, as the information was not essential for our analysis.

- Introduction general: The description of background on the Agulhas leakage is very detailed - maybe it could be condensed a bit to the main points? Likewise the section of ODS-induced ozone depletion is very detailed - could condense.

We shortened both paragraphs, as suggested. For the paragraph about the Agulhas leakage, we removed the detailed description of the impacts of Agulhas leakage changes. For the paragraph about ozone depletion, we removed the details about how ODS lead to ozone depletion and details about the early signs of ozone recovery.

- line 144 ff: Maybe rephrase - there have been many studies on past and future ozone and GHG effects (see e.g., WMO 2018), but what is new here is the in-cooperation of a (high-res) ocean model together with a CCM with interactive chemistry. This is stated in the following paragraph(s) (line 168) - consider to put this up front and combining the paragraphs to clarify.

We rephrased, shortened and moved the information about the new aspects of the study at the suggested location.

- line 197: It would be helpful to add the boundaries of the nests in one of the Figures.

The boundaries of the panels in Fig. 17 are the boundaries of the nest. We specified the limits of the nesting domain in the text and referred the reader to Fig. 17 and to the figure showing the nest imbedded in the global ocean in the paper by Schwarzkopf et al. (2019), which discusses the INALT10X nest in detail: "The nesting domain extends from 63°S to 10°N and from 70°W to 70°E (boundaries of Fig. 17, also depicted in Fig. 2 of Schwarzkopf et al. (2019))." (lines 242-243)

- line 242: Please include "quasi-geostrophic" to clarify that this is the QG version of the EP flux.

We added "quasi-geostrophic" to the sentence: "The quasi-geostropic Eliassen-Palm (EP) flux of wave activity, equivalent to the zonal mean of the vertical and meridional components of the Plumb flux, is defined as"

- line 250: Mostly the convention is to add a star (*) to the streamfunction as well to avoid confusion with the Eulerian mean streamfunction.

We added a star to the streamfunction, as suggested.

- line 287: consider reformulating to: "... CH4 cause an overall ozone increase, and N2O an overall ozone loss."

We reformulated as suggested.

- line 304: CMMI -> CCMI

We corrected the mistake, thank you for noticing it.

- line 345: "top of the tropospheric westerly jet ": in literature, this is often referred to as "upper flank of the subtropical jet", which is possibly a more specific phrasing.

We rephrased to "upper flank of the subtropical jet", as suggested.

- line 346: Are the effects linear (i.e. GHG+ODS = GHG&ODS ) ? In particular for dynamics that are highly non-linear, this cannot a priori expected. Maybe add a short note?

We obtain the ODS (GHG) effect by subtracting the end of the century mean in a given field of FixODS (FixGHG) from INTERACT_$O_3$. We obtain the total effect by subtracting the 2011-2030 mean from the 2080-2099 mean in INTERACT_$O_3$ or CMIP6_$O_3$. Therefore, we cannot expect that the sum of the ozone and GHG effects thus calculated would equal exactly the total effect.

- Fig. 4/5, line 359: Please add also climatological contours to Fig. 4 (as is done in Fig. 5), to simplify interpretation of the changes (are pattern attenuated or amplified?).

We added the contours as suggested.

- line 360: "due changes" -> "due to changes"

We corrected the mistake, thank you for pointing it out.

- line 432: consider rephrasing - the STJ does not extend into the stratosphere, but the wind changes up to the top of the stratosphere are rather changes in the polar vortex. For example consider: "... an acceleration of the STJ and enhanced westerly winds throughout the stratosphere"

We rephrased as suggested to make it clear that the positive change extends throughout the stratosphere and not the STJ.

- line 455: a planetary wave is better characterized by the wavenumber-decomposed EP flux rather than local wave activity changes, as it is planetary (and thus non-local) by nature (see major comment).

We included the EP flux divergence changes due to PW1 in the supplement (Fig. S3). The changes related to the higher wavenumbers are much weaker and mostly not significant.

- line 465: I strongly disagree with this interpretation. The "critical layer control" mechanism explained in Shepherd and McLandress is caused by a forcing-induced wind change (GHG warm the upper troposphere, thus inducing the wind changes by thermal wind balance), while it is not obvious why GHG forcing would lead to asymmetric changes in the zonal wind (GHG are well-mixed, so their purely radiative effect has to be zonally symmetric). Rather, the symmetric zonal wind changes are a signature of the planetary wave itself.

We removed the part that links the "critical layer control" to the changes in zonal wind and planetary wave propagation due to GHGs.

- line 477: In this case I agree with the interpretation: in the case of ozone forcing, there is a clear thermodynamic stating point (ozone changes drive temperature changes), that subsequently modifies the background wind through thermal wind balance; the PW propagation responds to the changed background winds. Again I strongly would recommend to include the (wavenumber decomposed) EP flux diagnostics instead of the 3-d wave activity flux in order to interpret the dynamical changes related to planetary waves.

We moved the total EP flux divergence change to the main manuscript and we included the EP flux divergence changes due to PW1 in the supplement (Fig. S3). The changes related to the higher wavenumbers are much weaker and mostly not significant.

- line 505: The introductory sentence of the section clarifies that only the residual circulation is analyzed here rather than the entire BDC - why not renaming the section to "Residual Circulation" ?

We changed the name from "The Brewer-Dobson circulation" to "The residual circulation and wave activity", as we combined sections 4.1.2 and 4.1.3.

- line 515: should read "north of ..", right ?

Yes, thank you. We corrected to "north of".

- line 543: I would recommend to show Figure S6 of the total EP flux divergence in the main paper, as it provides a more quantitative link between wave activity to the changes in residual circulation (see major comment).

The changes in the total EP flux divergence (former Fig. S6) were moved to the main manuscript and are now shown in Fig. 12.

- line 550: I agree with this interpretation. Could overlay wind changes and EP flux changes to highlight this effect more quantitatively.

We overlaid the changes in the zonal wind on the total EP flux divergence changes that are now included in the main manuscript (Fig. 12).

- line 571: So if the mechanism of GHG-induced residual circulation changes were to be investigated in more detail here, you would need to analyze EP fluxes / residual circulation outside the polar cap, right? I leave it to the authors if they want to go into more detail here.

The changes in the EP flux divergence shown in Figs. 12 and S3 are averaged over the latitude band 45S-80S. This was chosen as latitude-height plots of EP flux divergence changes in individual months showed that the changes occur in this band (Fig. R2). In addition, we show in Fig. 10 the downward mass flux and in Fig. S1 the changes in the mean residual streamfunction in the entire Southern Hemisphere. It is however not clear from any of these plots through which mechanism GHGs lead to the reported changes in wave activity.

[Figure]

Figure R2: Changes in the EP flux divergence (color shading) in CMIP6_$O_3$ (a), INTERACT_$O_3$ (b), only due to ozone recovery (c) only due to GHGs (d) and its current day climatology (contours) in m s$^{-1}$ day$^{-1}$. The stippling masks regions where the changes are not significant at the 95% confidence interval based on a two-tailed t-test.

- line 601 ff: consider moving this paragraph to the discussion section? This is rather speculative as it is hard to judge whether indeed the higher resolution ocean model leads to more realistic simulation of SST-related effects.

The paragraph was moved to the "Summary and discussion" section.

- line 671: I assume you refer to an uncoupled simulation? Please clarify.

The simulations used by Cheng et al., (2016, 2018) were performed with a coupled climate model (CCSM3.5), as stated.

- line 766: Do you see signs of this effect on the long-term trend in the simulations? Is it possible to quantify the effects of the Agulhas leakage on the Atlantic circulation trends?

We see the signs of increased (decreased) Agulhas leakage in the patterns of SST changes due to GHGs (ozone recovery) and we discuss the influence of the leakage in a new section about the ocean temperature, Sect. 4.2.4. Quantifying the temperature changes attributed to the change in Agulhas leakage is, however, complicated. As it is clear from Fig. 17, other factors also contributed to the temperature changes and it is not possible to clearly separate these different contributions. In addition, our high-resolution nest is restricted to the latitude bands 63°S-10°N, hence excludes many remote effects on the AMOC from the North Atlantic and the Southern Ocean.

- line 795: Possibly a scatter plot of wind stress versus ACC (similar to Fig. 14) could help to quantify this effect?

The scatter plot below shows that, while there is a general increase in the ACC transport with increasing wind stress, even decades of 20-30% higher wind stress are characterized by rather weak increases of up to 6% in ACC transport. This is a consequence of the eddy saturation effect. The black line represents a one-to-one increase ACC with wind stress. Also note the large spread of the ACC transport in the individual ensemble members (hollow circles), which is caused by a large spread in the density gradient across the ACC and which further complicates the interpretation of ACC transport changes.

[Figure]

Figure R3: Decadal changes in the ACC transport south of Africa versus decadal changes in wind stress over 45°S and 65°S and 30°E and 120°E (% change relative to 2014-2023). Each circle represents the change of each separate decade starting with 2014-2023 and ending with 2083-2094 relative to 2014-2023 in FixGHG (dark blue), INTERACT_$O_3$ (light blue), FixODS (light red) and CMIP6_$O_3$ (dark red). Filled circles denote ensemble means and small, hollow circles denote individual members. The black line depicts a one-to-one increase in the ACC with wind stress.

- Discussion / Summary general comment: The Discussion does in large parts summarize the results already; thus the Summary does duplicate the Discussion section to a large degree. I would recommend to either combine the two sections into one "Discussion and Summary" section, or avoid duplication by only focusing on the critical points in the discussion.

We removed the discussion section and moved the important part of the discussion to the summary section (now called "Summary and discussion"), or, where fitting, to the results section.

- Section 5.3: possibly combine with 5.2, as the wind, residual circulation and wave activity changes are closely coupled and some discussion is duplicated.

The changes in the residual circulation and wave activity are now discussed together in the combined "Summary and discussion" section.

-Discussion of ocean circulation changes: I wonder on the effects of having the "nests" of high-resolution ocean model - could the only partly high-resolved ocean induce artefacts, or would you expect even more compensation if the whole ocean was simulated at the resolution of nests? Consider adding a remark on this.

We restrict our analysis of the oceanic changes to the region of the high resolution nest, to avoid any issues that might arise from having different resolutions in the South Atlantic/western Indian Ocean and in the rest of the global ocean.

If the entire ocean would have an eddy-resolving resolution, the eddy saturation effect could result in even weaker ACC changes for the same wind stress change, as discussed in Sect. 5 at lines 1330-1337 in light of the results of Bishop et al. (2016). The sensitivity of the ACC to the ozone or GHG forcing is, however, further complicated by the fact that the ACC also responds to changes in the density gradient between the Weddell Sea and the South Atlantic, the latter being part of the nest while the former is not.

- line 980: consider citing recent studies that investigated those effects of interactive ozone.

We cited the study by Haase et al. (2020).

- line 990: remove () around Li et al 2016.

Thank you for pointing out the mistake.

---

## Author Response (AR2)

Dear Referees, Dear Editor,

Thank you for going over our manuscript once more and for your new helpful suggestion. Please see our response to these suggestions below. In addition to the modifications required by the referees, we noticed that $\phi$, a, and p in equations 1-5 of Section 2.2 were no longer defined after the previous revision round and we added their definitions. This happened as we moved the definition of the Plumb flux to the supplement during the last revision round.

**Referee #1**

The revisions made by the authors have address all the concerns raised in my initial review.

I have a minor suggestion for the authors to consider: In reading the original manuscript I did not pick up that the ocean included a higher resolution nested component, and even thought it is stated in the manuscript it is still as prominent as it could be as readers may also miss this. E.g., in the Introduction the statements on line 110-111 could be moved to line 104. This fact could also be included when discussing the ocean changes.

We emphasized the fact that we are using a high resolution ocean nest at lines 104-105 in the Introduction, 581-583 in the section about the Agulhas leakage changes and 747-748 at the beginning of the Summary and discussion section.

**Referee #2**

In the revised version of the manuscript by Ivanciu et al., the authors have considered all review comments appropriately and put great effort into incorporating the suggested changes and therewith improved the manuscript. All of my comments have been addressed satisfactory. The phrasing around the zonally asymmetric structure of stratospheric temperature changes does now reflect the underlying change in planetary waves very well, and in particular the revised section on dynamical changes (new section 4.1.2) nicely analyses the detected changes in a consistent manner.

Only minor comment I would have is that the combined Summary and Discussion section is now (still) rather long, making it difficult to extract the essential new findings and conclusions. The authors might want to consider to shorten some parts (e.g. the discussion around future surface / SAM impacts reads a bit like a review, and might go beyond what is required to discuss the results of the current study in the context of other literature).

We shortened the Summary and discussion section by eliminating details from the suggested paragraph, as well as from other paragraphs.